# Finite Sample Analysis of Average-Reward TD Learning and $Q$-Learning

**Sheng Zhang**[*]  **Zhe Zhang**[*]  **Siva Theja Maguluri**
The H. Milton Stewart School of Industrial and Systems Engineering
Georgia Institute of Technology
{shengzhang, jimmy_zhang, siva.theja}@gatech.edu

## Abstract

The focus of this paper is on sample complexity guarantees of average-reward reinforcement learning algorithms, which are known to be more challenging to study than their discounted-reward counterparts. To the best of our knowledge, we provide the first known finite sample guarantees using both constant and diminishing step sizes of (i) average-reward TD($\lambda$) with linear function approximation for policy evaluation and (ii) average-reward $Q$-learning in the tabular setting to find the optimal policy. A major challenge is that since the value functions are agnostic to an additive constant, the corresponding Bellman operators are no longer contraction mappings under any norm. We obtain the results for TD($\lambda$) by working in an appropriately defined subspace that ensures uniqueness of the solution. For $Q$-learning, we exploit the span seminorm contractive property of the Bellman operator, and construct a novel Lyapunov function obtained by infimal convolution of a generalized Moreau envelope and the indicator function of a set.

## 1 Introduction

The average-reward setting is a classical setting for formulating the goal in an infinite-horizon Markov decision process (MDP) [1]. The need to maximize the average reward has been demonstrated in many applications, including scheduling automatic guided vehicles [2], inventory management in supply chains [3], communication system control and routing [4], cooperative multi-robot learning [5] and queuing network control [6]. In these problems, the discounted-reward criterion usually leads to poor long-time performance since the system operates over an extended period of time and the goal is to optimize long-term behavior, whereas the discounted objective biases the optimal policy to choose actions that lead to high near-term performance.

Even though there is a well developed theory of average-reward MDPs [7, 8, 9], the theoretical understanding of average-reward RL methods is still quite limited. Most existing results are focused only on asymptotic convergence [10, 11, 12, 13]. The focus of this paper is to understand the sample efficiency. *How much data is required to guarantee a given level of accuracy?*

Recent literature obtains finite sample guarantees for discounted-reward TD learning and $Q$-learning algorithms by developing novel analytical techniques [14, 15, 16, 17, 18]. Such a study of average-reward RL algorithms is not undertaken. Analysis of average-reward RL algorithms is known to be more challenging to study than their discounted-reward counterparts. The key property that is exploited in the study of discounted-reward problems is the contraction property of the underlying Bellman operator. In the average-reward setting, such a contraction property does not hold under any norm, and the Bellman equation is known to have multiple fixed points.

---

[*]Equal contribution. Correspondence to Sheng Zhang <shengzhang@gatech.edu | shengzhangblog.com>.

35th Conference on Neural Information Processing Systems (NeurIPS 2021).

In this work, we take the first step toward understanding finite sample guarantees of (i) average-reward TD($\lambda$) with linear function approximation for policy evaluation, and (ii) average-reward tabular $Q$-learning in the synchronous setting for the control problem.

## 1.1 Contributions and Summary of Our Techniques

We establish the first finite sample convergence guarantees of average-reward TD($\lambda$) with linear function approximation and average-reward tabular $Q$-learning in the literature.

**TD($\lambda$) Results.** We study the average-reward TD($\lambda$) with linear function approximation under a general asynchronous update. We present finite sample bounds under both constant and diminishing step sizes. With a constant step-size, the iterates converge at an exponential rate to a small cylinder around the set of TD fixed points. With properly chosen diminishing step sizes, the mean-square distance of the iterates to the set of TD fixed points converges with an $\tilde{\mathcal{O}}\left(\frac{1}{T}\right)$ rate, and this leads to a sample complexity of $\tilde{\mathcal{O}}\left(\frac{1}{\epsilon^2}\right)$. Our sample complexity bound also suggests that an intermediate value of $\lambda$ yields the best performance. The dependence on the effective horizon plays a key role in the study of discounted-reward RL algorithms [19, 20]. There is no such effective horizon in average-reward problems, and the spectral gap of an appropriately defined matrix plays a key role instead.

**TD($\lambda$) Analysis.** A major challenge in the analysis is that the projected Bellman operator is not a contraction under any norm. Moreover, even though the projected Bellman equation can be written as a linear set of equations, they are underdetermined. So existing techniques [14, 15] are not directly applicable. Since the value function is unique up to an additive constant, we have a unique solution of the projected Bellman equation when restricted to an appropriately defined subspace. We exploit this property and work in this subspace, and use a quadratic Lyapunov function to obtain finite sample guarantees.

**$Q$-learning Results.** We consider a $J$-step synchronous $Q$-learning algorithm. We present finite sample error bounds for both constant and diminishing step sizes. The span of a vector is defined to be the difference between the maximum and minimum element. Since the optimal action-value function, $Q^*$, is agnostic up to an additive constant, we show that, with at most $O(1/\epsilon^2)$ samples, the expected span of the error $Q_T - Q^*$ converges to $\epsilon$ for both decreasing and constant step sizes.

**$Q$-learning Analysis.** While the corresponding Bellman operator is not a contraction under any norm, it is known to be a contraction under the span seminorm. The span seminorm can be interpreted as the $\ell_\infty$-distance to the subspace spanned by the all-ones vector. Finite sample bounds for stochastic approximation of $\ell_\infty$-norm contractive operators were obtained in [18] by using generalized Moreau envelop as a smooth Lyapunov function. Here, we generalize this approach and introduce a new Lyapunov function to study span seminorm contractive operators. Our Lyapunov function is obtained by applying an infimal convolution with respect to an indicator function to the generalized Moreau envelop used in [18].

## 1.2 Related Literature

**Average-Reward MDP.** There is an extensive body of literature on average-reward MDPs. Several authors have made early contributions to average-reward problems [21, 7, 8, 22, 23]. There are well known dynamic programming algorithms for finding optimal policies such as policy iteration [7] and value iteration [24]. However, these algorithms require complete knowledge of the MDP, and are also computationally intractable in large state spaces [25].

**Average-Reward Policy Evaluation.** Tsitsiklis and Van Roy [10] proved the asymptotic convergence of the average-reward TD($\lambda$) with linear function approximation, and provided approximation error bounds. Yu and Bertsekas [26] proved the asymptotic convergence of the average-reward LSPE($\lambda$), and provided the rate of convergence for constant step size. Both TD($\lambda$) with linear function approximation and LSPE($\lambda$) aim to solve the same projected Bellman equation. However, TD($\lambda$) is based on stochastic approximation while LSPE($\lambda$) is based on least squares. In addition, the papers above assumed that the set of basis functions are independent of the all-ones vector, which apparently does not hold in the tabular setting. We do not require such a restrictive assumption in this paper. Recent work has also established the asymptotic convergence of the off-policy average-reward TD learning algorithm in the tabular setting [12], and finite sample guarantees of average-reward gradient

TD algorithms with linear function approximation [13]. Note that, [13] needed either the restrictive assumption aforementioned or the ridge regularization to obtain the finite sample results.

**Average-Reward Control.** The earliest control algorithms were those introduced by Schwartz [27] and Singh [28] without convergence proofs. The first provably convergent algorithms are RVI $Q$-learning and SSP $Q$-learning, introduced by Abounadi, Bertsekas, and Borkar [11]. SSP $Q$-learning and the algorithm introduced later by Gosavi [29] are limited to MDPs with a special state that is recurrent under all stationary policies, whereas RVI $Q$-learning is convergent for more general MDPs. Recently, Wan et al.[12] introduced an algorithm without a reference function, which is needed in RVI $Q$-learning, and proved its asymptotic convergence with the techniques which are a slight generalization of those in [11]. To the best of our knowledge, our paper is the first work in the literature that studies the finite sample guarantees of a general average-reward $Q$-learning algorithm.

**Stochastic Approximation.** Many RL algorithms can be viewed through the lens of stochastic approximation (SA). There is a well developed asymptotic theory of SA [30, 31, 32]. The ODE method is a dominant approach used in most asymptotic convergence proofs in RL [33]. However, this is a coarse tool, since it is not able to generate insight into an algorithm's sensitivity to noise in the system and step-size choices. Driven by the interest in finite sample guarantees of RL algorithms, recent years have witnessed a focus shifted from asymptotic analysis to non-asymptotic analysis of SA schemes. For example, a finite-time bound for linear SA was given in [15], which leads to finite-time bounds for asynchronous TD learning. [17] provided a finite-time analysis of asynchronous nonlinear SA, which yields finite-time bounds for asynchronous $Q$-learning.

**Others.** There are other related papers which are beyond the scope of the present paper. For instance, there is a line of work [34, 35, 36] on regret guarantees, which is a different focus compared to our work, for learning in average-reward MDPs. In addition, there are RL methods based on linear programming [37, 38], or learning automata [39, 40].

## 2  The Average-Reward Problem Setting

We consider an infinite-horizon average-reward MDP described by $(\mathcal{S}, \mathcal{A}, \mathcal{R}, p)$, where $\mathcal{S} = \{1, 2, \cdots, |\mathcal{S}|\}$ is a finite state space, $\mathcal{A} = \{1, 2, \cdots, |\mathcal{A}|\}$ is a finite action space, $\mathcal{R} : \mathcal{S} \times \mathcal{A} \to [0, 1]$ is the reward function, and $p : \mathcal{S} \times \mathcal{S} \times \mathcal{A} \to [0, 1]$ is the transition dynamics of the environment. An agent interacts with the environment according to the following protocol: at each time step $t = 0, 1, 2, \cdots$, the agent is in a state $S_t \in \mathcal{S}$ and selects an action $A_t \in \mathcal{A}$, then receives from the environment an immediate reward $\mathcal{R}(S_t, A_t)$ and the next state $S_{t+1}$ which is a sample drawn from $p(\cdot|S_t, A_t)$. The average reward of a deterministic stationary policy $\mu : \mathcal{S} \to \mathcal{A}$ starting from state $s \in \mathcal{S}$ is defined as

$$r^\mu(s) := \liminf_{T \to \infty} \frac{1}{T} \mathbb{E}\left[\sum_{t=0}^{T-1} \mathcal{R}(S_t, \mu(S_t))|S_0 = s\right]. \tag{2.1}$$

Let $r^*(s) := \sup_{\mu \in \mathcal{M}} r^\mu(s)$, where $\mathcal{M}$ is the set of deterministic stationary policies. A policy $\mu^* \in \mathcal{M}$ is said to be optimal if it satisfies $r^{\mu^*}(s) = r^*(s)$ for all $s \in \mathcal{S}$.

## 3  Policy Evaluation Algorithm: TD Learning

### 3.1  Problem Formulation

We consider the problem of evaluating a given policy $\mu \in \mathcal{M}$ when the data is generated by applying the policy $\mu$ in the MDP. Since the system is an induced Markov reward process (MRP), for simplicity, we employ the notation $\mathcal{R}(i) := \mathcal{R}(i, \mu(i))$ for rewards, and $P(i, j) := p(j|i, \mu(i))$ for transition probabilities. We make the following standard assumption to ensure the existence and uniqueness of a stationary distribution $\pi := [\pi_1, \cdots, \pi_{|\mathcal{S}|}]^\top$.

**Assumption 1.** *The Markov chain associated with $P$ is irreducible and aperiodic*[2].

---

[2]It is a standard assumption in studying convergence of TD learning algorithms with linear function approximation [41, 10, 14] and guarantees that all states are visited an infinite number of times during an infinitely long trajectory.

Notice that $\pi$ satisfies $\pi^\top P = \pi^\top$, with $\pi_i > 0$ for all $i \in \mathcal{S}$. Let $\mathbb{E}_\pi[\cdot]$ denote expectation with respect to $\pi$ and define $D = \text{diag}(\pi_1, \cdots, \pi_{|\mathcal{S}|}) \in \mathbb{R}^{|\mathcal{S}| \times |\mathcal{S}|}$. It is easy to see that $\langle x, y \rangle_D := x^\top D y$ is a $D$-weighted inner product and we denote its induced norm by $\|x\|_D := \sqrt{x^\top D x}$.

Under Assumption 1, the average reward in (2.1) satisfies $r^\mu(s) = r(\mu) := \pi^\top \mathcal{R}$ for all $s \in \mathcal{S}$. A differential value function $v : \mathcal{S} \to \mathbb{R}$ for policy $\mu$ satisfies the Bellman equation $Tv = v$, where the Bellman operator $T : \mathbb{R}^{|\mathcal{S}|} \to \mathbb{R}^{|\mathcal{S}|}$ is defined by $Tv := \mathcal{R} - r(\mu)e + Pv$. Here, $e \in \mathbb{R}^{|\mathcal{S}|}$ is the all-ones vector. Under Assumption 1, it is known that the set of differential value functions takes the form $\{v^\mu + ce | c \in \mathbb{R}\}$, where $v^\mu : \mathcal{S} \to \mathbb{R}$, known as the *basic* differential value function, is given by $v^\mu := \sum_{t=0}^\infty P^t (\mathcal{R} - r(\mu)e)$.

Most modern applications have large state spaces, so due to the curse of dimensionality, exact value function learning may be intractable. To mitigate this, we consider a linear function approximation $V_\theta(i) = \phi(i)^\top \theta$ to differential value functions, where $\phi(i) := [\phi_1(i), \cdots, \phi_d(i)]^\top \in \mathbb{R}^d$ is the feature vector for state $i \in \mathcal{S}$ and $\theta \in \mathbb{R}^d$ is a tunable parameter vector. Here, $\{\phi_k : \mathcal{S} \to \mathbb{R} | k = 1, 2, \cdots, d\}$ is a set of $d$ basis functions to be viewed as vectors of dimension $|\mathcal{S}|$. With this notation, $V_\theta$ can be expressed compactly in the form $V_\theta = \Phi\theta$, where $\Phi$ is an $|\mathcal{S}| \times d$ matrix whose $k$-th column is $\phi_k$. We assume that $\Phi$ has full column rank; that is, the basis functions $\{\phi_k | k = 1, 2, \cdots, d\}$ are linearly independent. This results in no loss of generality because if some basis function $\phi_k$ is a linear combination of the others, it can be eliminated without changing the power of the approximation architecture. Additionally, we assume that $\|\phi(i)\|_2 \leq 1$ for all $i \in \mathcal{S}$, which can be ensured through feature normalization.

## 3.2 Average-Reward TD($\lambda$)

---

**Algorithm 1:** TD($\lambda$) with linear function approximation

---

**Input** : initial guess $\bar{r}_0$ and $\theta_0$, basis functions $\{\phi_k\}_{k=1}^d$, step-size sequence $\{\beta_t\}_{t \in \mathbb{N}}$ and positive constant $c_\alpha$.
Initialize: $z_{-1} = 0$, $\lambda \in [0, 1)$.
**for** $t = 0, 1, \ldots$ **do**

> Observe tuple: $O_t = (s_t, \mathcal{R}(s_t), s_{t+1})$
> Get TD error: $\delta_t(\theta_t) = \mathcal{R}(s_t) - \bar{r}_t + \phi(s_{t+1})^\top \theta_t - \phi(s_t)^\top \theta_t$
> Update eligibility trace: $z_t = \lambda z_{t-1} + \phi(s_t)$
> Update average-reward estimate: $\bar{r}_{t+1} = \bar{r}_t + c_\alpha \beta_t (\mathcal{R}(s_t) - \bar{r}_t)$
> Update parameter vector: $\theta_{t+1} = \theta_t + \beta_t \delta_t(\theta_t) z_t$

**end**

---

We study the average-reward TD learning with eligibility traces [10], denoted by TD($\lambda$) and parameterized by $\lambda \in [0, 1)$. We consider the Markov chain observation model, where the observed tuples used by TD($\lambda$) are gathered from a single trajectory of the MRP. At every time step $t$, the algorithm observes one data tuple $O_t := (s_t, \mathcal{R}(s_t), s_{t+1})$ consisting of the current state, the current reward and the next state. Suppose that at some time $t$, the current value of the parameter vector $\theta$ is $\theta_t$, and we have a scalar estimate $\bar{r}_t$ of the average reward $r(\mu)$, we define the TD error $\delta_t(\theta_t)$ corresponding to the transition from $s_t$ to $s_{t+1}$ as $\delta_t(\theta_t) := \mathcal{R}(s_t) - \bar{r}_t + \phi(s_{t+1})^\top \theta_t - \phi(s_t)^\top \theta_t$. TD($\lambda$) updates $\bar{r}_t$ and $\theta_t$ as follows:

$$\bar{r}_{t+1} = \bar{r}_t + \alpha_t(\mathcal{R}(s_t) - \bar{r}_t) \text{ and } \theta_{t+1} = \theta_t + \beta_t \delta_t(\theta_t) z_t, \tag{3.1}$$

where $\alpha_t$ and $\beta_t$ are scalar step sizes, and the vector $z_t := \sum_{k=0}^t \lambda^{t-k} \phi(s_k)$ is called the eligibility trace.

In this work, we focus on the single time-scale variant of TD($\lambda$) presented in Algorithm 1, that is, we assume that there exists a constant $c_\alpha > 0$ such that $\alpha_t = c_\alpha \beta_t$ for all $t$. In order to represent TD($\lambda$) in a compact form, we construct a process $X_t := (s_t, s_{t+1}, z_t)$. It is easy to see that $\{X_t\}$ is a Markov chain with an infinite state space. If we let $\Theta_t := \begin{bmatrix} \bar{r}_t \\ \theta_t \end{bmatrix}$, the TD($\lambda$) updates (3.1) can be expressed compactly as

$$\Theta_{t+1} = \Theta_t + \beta_t [A(X_t)\Theta_t + b(X_t)], \tag{3.2}$$

where

$$A(X_t) = \begin{bmatrix} -c_\alpha & 0 \\ -z_t & z_t \left( \phi(s_{t+1})^\top - \phi(s_t)^\top \right) \end{bmatrix} \text{ and } b(X_t) = \begin{bmatrix} c_\alpha \mathcal{R}(s_t) \\ \mathcal{R}(s_t) z_t \end{bmatrix}.$$

**Non-uniqueness of TD($\lambda$) limit point.** For any $m = 0, 1, \cdots$, the $m$-step Bellman operator is given by

$$T^m v = \sum_{t=0}^{m} P^t \left( \mathcal{R} - r(\mu)e \right) + P^{m+1}v.$$

The asymptotic properties of TD($\lambda$) are closely tied to the $\lambda$-weighted version of the $m$-step Bellman operator. Define the averaged Bellman operator

$$T^{(\lambda)}v := (1 - \lambda) \sum_{m=0}^{\infty} \lambda^m T^{m+1}v.$$

Note that the set of differential value functions is also the fixed points of $T^{(\lambda)}$; see Lemma 3 in [10] for a proof.

We denote by $A := \mathbb{E}_\pi[A(X_t)]$ and $b := \mathbb{E}_\pi[b(X_t)]$ the steady-state expectations of $A(X_t)$ and $b(X_t)$, respectively. Stochastic approximation theory [32] shows that the asymptotic behavior of the sequence $\{\Theta_t\}$ generated by (3.2) is closely linked with the corresponding ordinary differential equation (ODE) $\dot{\Theta}_t = A\Theta_t + b$, and the limit of $\Theta_t$, denoted by $\Theta_\infty = \begin{bmatrix} \bar{r}_\infty \\ \theta_\infty \end{bmatrix}$ if exists, is an equilibrium point of the ODE, i.e.,

$$A\Theta_\infty + b = 0. \tag{3.3}$$

Therefore, $\bar{r}_\infty = r(\mu)$, and $\theta_\infty$ is a solution of the projected Bellman equation

$$\Phi\theta = \Pi_{D, W_\Phi} T^{(\lambda)} \Phi\theta, \tag{3.4}$$

where $\Pi_{D, W_\Phi} := \Phi \left( \Phi^\top D\Phi \right)^{-1} \Phi^\top D$ is the projection matrix onto the column space $W_\Phi := \{\Phi\theta | \theta \in \mathbb{R}^d\}$ of $\Phi$ with respect to the norm $\|\cdot\|_D$. It is worth noting that if $e \in W_\Phi$, then $\Pi_{D, W_\Phi} T^{(\lambda)}$ has multiple fixed points, since any scalar multiple of $e$ when added to a fixed point of $\Pi_{D, W_\Phi} T^{(\lambda)}$ would also be a fixed point. For example, in the tabular case where $\Phi = I$, Eq. (3.4) would become $\theta = T^{(\lambda)}\theta$, of which the set of differential value functions are solutions.

### 3.3 Finite-Time Bounds for Average-Reward TD($\lambda$)

Before we present the finite-time bounds on the performance of TD($\lambda$) with Markovian observation noise, we illustrate the key ideas, which are inspired by [15]. The detailed proof can be found in Appendix A.3.

We study the drift of an appropriately chosen Lyapunov function to obtain an upper bound on the mean-square error. We define the subspace $S_{\Phi, e}$ as

$$S_{\Phi, e} := \text{span}\left(\{\theta | \Phi\theta = e\}\right) = \begin{cases} \{0\}, & \text{if } e \notin W_\Phi, \\ \{c\theta_e | c \in \mathbb{R}\}, & \text{if } e \in W_\Phi \text{ and } \Phi\theta_e = e. \end{cases}$$

Let $E$ be the orthogonal complement of $S_{\Phi, e}$. We can interpret $E$ as the set of equivalent classes with the equivalence relation $\sim$ on $\mathbb{R}^d$ defined by $\theta_1 \sim \theta_2$ if and only if $\theta_1 - \theta_2$ is in $S_{\Phi, e}$. The following lemma characterizes the set of TD($\lambda$) fixed points; see Appendix A.1 for a proof.

**Lemma 1.** *Under Assumption 1, the fixed points of the projected Bellman equation (3.4) are*

$$\mathcal{L} := \theta^* + S_{\Phi, e} = \begin{cases} \{\theta^*\}, & \text{if } e \notin W_\Phi, \\ \{\theta^* + c\theta_e | c \in \mathbb{R}\}, & \text{if } e \in W_\Phi \text{ and } \Phi\theta_e = e, \end{cases}$$

*where $\theta^* \in E$ is a unique solution to the equation $\Phi\theta = \Pi_{D, W_{\Phi, E}} T^{(\lambda)} \Phi\theta$. Here, $\Pi_{D, W_{\Phi, E}}(\cdot)$ is the projection operator onto the subspace $W_{\Phi, E} := \{\Phi\theta | \theta \in E\}$ with respect to the norm $\|\cdot\|_D$*

**Remark 1.** *In the case where $e \notin W_\Phi$, the projected Bellman equation (3.4) has a unique fixed point $\theta^*$. This is why prior work requires that $e$ does not belong to the column space of $\Phi$.*

We consider the Lyapunov function $\Phi(\bar{r}, \theta) := (\bar{r} - r(\mu))^2 + \|\Pi_{2,E}(\theta - \theta^*)\|_2^2$. Here, $\Pi_{2,E}$ is the projection onto the subspace $E$ with respect to the 2-norm $\|\cdot\|_2$. Note that $\|\Pi_{2,E}(\theta - \theta^*)\|_2^2$ measures the distance of $\theta$ to the set of TD($\lambda$) fixed points. The following Lemma establishes that the matrix $A$ in (3.3) is negative definite over the subspace $\mathbb{R} \times E$ for a sufficiently large $c_\alpha$. The proof is presented in Appendix A.2. With this result, we can show that the Lyapunov function $\Phi(\bar{r}, \theta)$ has a one-time-step negative drift.

**Lemma 2.** *Under Assumption 1, we have*

$$\Delta := \min_{\|\theta\|_2 = 1, \theta \in E} \theta^\top \Phi^\top D \left( I - P^{(\lambda)} \right) \Phi\theta > 0, \tag{3.5}$$

*where $P^{(\lambda)} = (1 - \lambda) \sum_{m=0}^\infty \lambda^m P^{m+1}$. Furthermore, when $c_\alpha \geq \Delta + \sqrt{\frac{1}{\Delta^2(1-\lambda)^4} - \frac{1}{(1-\lambda)^2}}$, we have*

$$\min_{\|\Theta\|_2 = 1, \Theta \in \mathbb{R} \times E} -\Theta^\top A \Theta \geq \frac{\Delta}{2}.$$

To handle the Markovian noise, we use the conditioning argument along with the geometric mixing of the underlying Markov chain $\{X_t\}$. Thus, we consider the following definition of the mixing time of a Markov chain.

**Definition 1.** *Given a positive constant $\epsilon$, we define $\tau(\epsilon) \geq 1$ to be the mixing time of a Markov chain $\{X_t\}$ such that*

$$\|\mathbb{E}[A(X_t)|X_0] - A\|_2 \leq \epsilon \text{ and } \|\mathbb{E}[b(X_t)|X_0] - b\|_2 \leq \epsilon, \quad \text{for any } X_0 \text{ and } t \geq \tau(\epsilon)$$

The following lemma establishes that the expectations of $A(X_t)$ and $b(X_t)$ converge to their steady-state values at a geometric rate. See Lemma 6.7 in [42] for a proof.

**Lemma 3.** *Under Assumption 1, the Markov chain $\{X_t\}$ has a geometric mixing time, i.e., there exists a constant $K \geq 1$ such that given a small positive constant $\epsilon$ we have $\tau(\epsilon) \leq K \ln\left(\frac{1}{\epsilon}\right)$.*

We now state two finite-time bounds on the performance of TD($\lambda$). Part (a) studies TD($\lambda$) applied with sufficiently small constant step-size, which is common in practice. In this case, the iterates $\theta_t$ will never converge to any TD($\lambda$) fixed point due to the noise variance, but our result shows that the expected distance of $\theta_t$ to the set of TD($\lambda$) fixed points decreases at an exponential rate below some level that depends on the choice of step-size. Part (b) attains an $\tilde{\mathcal{O}}\left(\frac{1}{T}\right)$ convergence rate to some TD($\lambda$) fixed point with a carefully chosen decaying step-size sequence.

**Theorem 1.** *Consider iterates $\{(\bar{r}_t, \theta_t)\}$ generated by Algorithm 1 with Assumption 1 and $c_\alpha \geq \Delta + \sqrt{\frac{1}{\Delta^2(1-\lambda)^4} - \frac{1}{(1-\lambda)^2}}$. Let $\xi_1 := \left(2\sqrt{\bar{r}_0^2 + \|\theta_0\|_2^2} + \sqrt{r(\mu)^2 + \|\theta^*\|_2^2} + 1\right)^2$ and $\xi_2 := 228\eta^2 \left(\sqrt{r(\mu)^2 + \|\theta^*\|_2^2} + 1\right)^2$, where $\eta := \sqrt{c_\alpha^2 + \frac{5}{(1-\lambda)^2}}$.*

*(a) Let $\beta_t = \beta$ for all $t$, where positive constant $\beta$ is properly chosen such that $\Delta\beta < 2$ and $\beta\tau(\beta) \leq \min\{\frac{1}{4\eta}, \frac{\Delta}{228\eta^2}\}$. Then, for all $T \geq \tau(\beta)$, we have*

$$\mathbb{E}\left[(\bar{r}_T - r(\mu))^2 + \|\Pi_{2,E}(\theta_T - \theta^*)\|_2^2\right] \leq \xi_1 \left(1 - \frac{\Delta}{2}\beta\right)^{T - \tau(\beta)} + \xi_2 \frac{\beta\tau(\beta)}{\Delta}.$$

*(b) Let $\beta_t = \frac{c_1}{t + c_2}$ where positive constants $c_1$ and $c_2$ are properly chosen such that $2 < \Delta c_1 < 2c_2$ and there exists a smallest positive integer $t^*$ such that $\sum_{k=0}^{t^*-1} \beta_k \leq \frac{1}{2\eta}$, and for all $t \geq t^*$, $\sum_{k=t-\tau(\beta_t)}^{t-1} \beta_k \leq \min\{\frac{1}{4\eta}, \frac{\Delta}{228\eta^2}\}$. Then, for all $T \geq t^*$, we have*

$$\mathbb{E}\left[(\bar{r}_T - r(\mu))^2 + \|\Pi_{2,E}(\theta_T - \theta^*)\|_2^2\right] \leq \xi_1 \left(\frac{t^* + c_2}{T + c_2}\right)^{\frac{\Delta c_1}{2}} + \frac{8eKc_1^2\xi_2}{\Delta c_1 - 2} \frac{\ln(T + c_2) - \ln(c_1)}{T + c_2 + 1}.$$

Therefore, with an appropriate decaying step sizes suggested in Theorem 1(b), the following sample complexity of Algorithm 1 can be obtained.

**Corollary 1.** *To find a pair $(\bar{r}, \theta)$ with $\mathbb{E}[|\bar{r} - r(\mu)|] \leq \epsilon$ and $\mathbb{E}\left[\|\Pi_{2,E}(\theta - \theta^*)\|_2\right] \leq \epsilon$, Algorithm 1 requires at most the following number of samples:*

$$\tilde{\mathcal{O}}\left(\frac{K \log\left(\frac{1}{\Delta}\right) \|\theta^*\|_2^2}{\Delta^4 (1-\lambda)^4 \epsilon^2}\right),$$

*where $K$ is the mixing time constant defined in Lemma 3.*

**Remark 2.** *Since $\Delta$ defined in (3.5) is a non-decreasing function of $\lambda$, the sample complexity in Corollary 1 implies that the optimal $\lambda$ should be neither too large nor too small.*

### 3.4 Approximation Error

As we are satisfied with an approximation of any differential value function, we define the approximation error, $\inf_{c \in \mathbb{R}} \|\Phi\theta^* - (v^\mu + ce)\|_D$, as the infimum of the $D$-weighted Euclidean distance of $\Phi\theta^*$ to the set of differential value functions. Following the similar arguments from the proof of Theorem 3 in [10], we obtain the following approximation error bound,

$$\inf_{c \in \mathbb{R}} \|\Phi\theta^* - (v^\mu + ce)\|_D \leq \frac{1}{\sqrt{1 - c_\lambda^2}} \inf_{\theta \in \mathbb{R}^d, c \in \mathbb{R}} \|\Phi\theta - (v^\mu + ce)\|_D, \tag{3.6}$$

where the constant $c_\lambda$ is in $[0, 1)$ for any $\lambda \in [0, 1)$ and goes to 0 as $\lambda \to 1$. Note that the term $\inf_{\theta \in \mathbb{R}^d, c \in \mathbb{R}} \|\Phi\theta - (v^\mu + ce)\|_D$ is the minimal error possible given our approximation architecture, and becomes zero if our approximation architecture is able to represent exactly some differential value function. In particular, under the tabular setting, since any differential value function has exact representation, we have $\inf_{c \in \mathbb{R}} \|\Phi\theta^* - (v^\mu + ce)\|_D = 0$.

## 4 Control Algorithm: $Q$-learning

### 4.1 Problem Formulation

We consider the problem of finding an optimal policy $\mu^* \in \mathcal{M}$ under the following unichain assumption (see Section 8.4 in [9] for details).

**Assumption 2.** *An MDP is called unichain if the induced Markov chain consists of a single recurrent class plus a possibly empty set of transient states for any deterministic stationary policy.*

Under Assumption 2, standard MDP theory [9] shows that there exist a unique $r^* \in \mathbb{R}$ such that $r^*(s) = r^*$ for all $s \in \mathcal{S}$, and a unique $Q^* : \mathcal{S} \times \mathcal{A} \to \mathbb{R}$ up to an additive constant, such that the following Bellman optimality equation holds for all state-action pairs $(s, a) \in \mathcal{S} \times \mathcal{A}$:

$$Q^*(s, a) = \mathcal{R}(s, a) + \sum_{s' \in \mathcal{S}} p(s'|s, a) \max_{a' \in \mathcal{A}} Q^*(s', a') - r^*. \tag{4.1}$$

The optimal policy $\mu^*$ is then obtained by $\mu^*(s) := \arg\max_{a \in \mathcal{A}} Q^*(s, a)$. If we define $\bar{E} := \{ce | c \in \mathbb{R}\}$ as the subspace spanned by the all-ones vector $e \in \mathbb{R}^{|\mathcal{S}| \times |\mathcal{A}|}$ and denote the Bellman operator $H$ by

$$H(Q)(s, a) = \mathcal{R}(s, a) + \sum_{s' \in \mathcal{S}} p(s'|s, a) \max_{a' \in \mathcal{A}} Q^*(s', a'), \quad \forall (s, a) \in \mathcal{S} \times \mathcal{A},$$

then (4.1) can be rewritten as a set inclusion condition:

$$Q^* - H(Q^*) \in \bar{E}. \tag{4.2}$$

Importantly, by observing that the operator $H$ is indifferent to constant shifting, i.e., $H(Q + ce) = H(Q) + ce$, we can view all $Q$ constant shifts, $Q_{\bar{E}} := \{Q + ce : c \in \mathbb{R}\}$, as an equivalent class and interpret (4.2) as a fixed-set equation:

$$Q_{\bar{E}}^* = H(Q_{\bar{E}}^*). \tag{4.3}$$

Next we propose an SA algorithm to solve (4.3) by iteratively updating some "representative" of the underlying equivalent class.

---

**Algorithm 2:** $J$-step Synchronous $Q$-learning

---

**Input** : initial guess $Q_0$, step-size sequence $\{\eta_t\}_{t\in\mathbb{N}}$, offset function $f : \mathbb{R}^{|\mathcal{S}|\times|\mathcal{A}|} \to \mathbb{R}$.

**for** $t = 0, 1, \ldots$ **do**

    Compute the $Q_t$-improving policy $\mu_t(s) := \arg\max_a Q_t(s, a)$ for all $s \in \mathcal{S}$.

    **for** $(s, a) \in \mathcal{S} \times \mathcal{A}$ **do**

        Sample $s^1 \sim p(\cdot|s, a), s^2 \sim p(\cdot|s^1, \mu_t(s^1)), \ldots, s^J \sim p(\cdot|s^{J-1}, \mu_t(s^{J-1}))$.

        Compute $Q_{t+1}(s, a) \leftarrow Q_t(s, a) +$

        $\eta_t \left( \mathcal{R}(s, a) + \sum_{j=1}^{J-1} \mathcal{R}(s^j, \mu_t(s^j)) + Q_t(s^J, \mu_t(s^J)) - Q_t(s, a) - f(Q_t) \right).$

    **end**

**end**

---

## 4.2 Synchronous $Q$-learning

Given the sample Bellman operator $\hat{H}$ defined by

$$\hat{H}(Q)(s, a) := \mathcal{R}(s, a) + \max_{a'\in\mathcal{A}} Q(s', a'), \; s' \sim p(\cdot|s, a), \quad \forall (s, a) \in \mathcal{S} \times \mathcal{A},$$

an SA algorithm for solving (4.3) is

$$Q_{t+1} := Q_t + \eta_t \left( \hat{H}(Q_t) - Q_t \right). \tag{4.4}$$

It might be necessary, sometimes, to apply $H$ to $Q$ for $J$ steps before updating $Q$. More specifically, if we denote by $\mu_Q(s) := \arg\max_a Q(s, a)$ the $Q$-improving policy, the $J$-step Bellman operator and the $J$-step fixed-set equation are

$$
\begin{aligned}
H^J(Q)(s, a) := \mathcal{R}(s, a) + \mathbb{E}_{s^1 \sim P(\cdot|s,a),\ldots,s^J \sim P(\cdot|s^{J-1}, \mu_Q(s^{J-1}))} \big[ \\
R(s^1, \mu_Q(s^1)) + \ldots + R(s^{J-1}, \mu_Q(s^{J-1})) + Q(\mu^J, \mu_Q(s^J)) \big],
\end{aligned} \tag{4.5}
$$

$$H^J(Q^*_{\bar{E}}) = Q^*_{\bar{E}}. \tag{4.6}$$

So the $J$-step SA algorithm is the same as (4.4) except for a $J$-step sample Bellman operator,

$$\hat{H}^J(Q)(s, a) := \mathcal{R}(s, a) + R(s^1, \mu_Q(s^1)) + \ldots + R(s^{J-1}, \mu_Q(s^{J-1})) + Q(\mu^J, \mu_Q(s^J)),$$

where $s^1 \sim p(\cdot|s, a), \ldots, s^J \sim p(\cdot|s^{J-1}, \mu_Q(s^{J-1}))$. The complete algorithm is presented in Algorithm 2.

**Remark 3.** *(1) The SA algorithm in (4.4) is a special case of Algorithm 2 with $J = 1$. (2) The solution to the $J$-step fixed-set equation (4.6) is the same for all $J \geq 1$. (3) An extra offset function $f(Q)$ is included to ensure numerical stability (see Section 2.2 in [11]). If $J = 1$ and the offset function $f$ satisfies Assumption 2.2 in [11], Algorithm 2 recovers the RVI $Q$-learning algorithm.*

## 4.3 Finite-Time Analysis

Now we seek to establish finite-time convergence guarantees of Algorithm 2. Since the $Q$-improving policy and its suboptimality gap are the same for any $Q \in Q_{\bar{E}}$, we can measure Algorithm 2's progress by

$$\|Q - Q^*\|_{\alpha,\mathrm{sp}} := \min_{c\in\mathbb{R}} \|(Q - Q^*) - ce)\|_{\alpha}, \tag{4.7}$$

for some norm $\|\cdot\|_{\alpha}$. For example, the span seminorm is equivalent to $\|\cdot\|_{\infty,\mathrm{sp}}$. Additionally, we need to make a span contraction assumption similar to Theorem 8.5.2 in [9].

**Assumption 3.** *The $J$-step Bellman operator $H^J$ is a span contraction for some $J \geq 1$, i.e., there exists a $\gamma \in [0, 1)$ such that for any $Q_1$ and $Q_2$ defined on $\mathcal{S} \times \mathcal{A}$,*

$$\left\| H^J(Q_1) - H^J(Q_2) \right\|_{\infty,\mathrm{sp}} \leq \gamma \left\| Q_1 - Q_2 \right\|_{\infty,\mathrm{sp}}.$$

Such an assumption is not restrictive. A lower bound for $1 - \gamma$ is the minimum probability of any two deterministic stationary Markov policies starting from any pair of states ending in the same state

in $J$ steps [9]. Thus, under an unichain assumption, $\gamma < 1$ can be guaranteed for $J = |S|$ if we apply the aperiodic transformation in Section 8.5.4 of [9] to ensure a non-zero probability of all states in a single recurrent class in $|S|$ steps.

We sketch the outline of our Lyaponuv convergence proof and leave the details to Appendix B. First, we construct a Lyaponov function. The key insight is that the span seminorm $\|\cdot\|_{\infty,\mathrm{sp}}$ can be interpreted as the infimal convolution of the $\ell_\infty$-norm $\|\cdot\|_\infty$ and the indicator function of $\bar{E}$, i.e.,

$$\|x\|_{\infty,\mathrm{sp}} = (\|\cdot\|_\infty \,\square\, \delta_{\bar{E}})(x) := \inf_y \|x - y\|_\infty + \delta_{\bar{E}}(y), \quad \forall x, \tag{4.8}$$

where $\delta_{\bar{E}}(x) := \begin{cases} 0, & x \in \bar{E}, \\ \infty, & \text{otherwise.} \end{cases}$

As illustrated by Lemma 4 in Appendix B.1, the infimal convolution operation has many desirable properties. For example, it is commutative, associative, convexity-preserving and smoothness-preserving. These nice properties allow us to design a Lyaponov function $M_{\bar{E}}$ for the equivalent classes by tweaking the usual Lyaponov function $M$ as follows,

$$M_{\bar{E}}(Q) := (M \square \delta_{\bar{E}})(Q). \tag{4.9}$$

In particular, since Assumption 3 implies only span contraction, we utilize the smoothed Lyaponov function proposed in [18] for $\ell_\infty$-norm contraction, i.e., $M(Q) := \frac{1}{2}(\|\cdot\|_\infty^2 \,\square\, \frac{1}{u}\,\|\cdot\|_p^2)(Q)$ with $p := 4\log(|S||A|)$ and $u := (1/2 + 1/(2\gamma))^2 - 1$. We show $M_{\bar{E}}(\cdot)$ is a uniform approximation of $\|\cdot\|_{\infty,\mathrm{sp}}^2$:

$$(1 + (1 + 1/\gamma)^2/4\sqrt{e})M_{\bar{E}}(Q) \le \tfrac{1}{2}\|Q\|_{\infty,\mathrm{sp}}^2 \le (1 + (1 + 1/\gamma)^2/4)M_{\bar{E}}(Q), \quad \forall Q,$$

and it is smooth with respect to $\|\cdot\|_{p,\mathrm{sp}}$:

$$\begin{aligned} M_{\bar{E}}(Q_{t+1} - Q^*) \le{}& M_{\bar{E}}(Q_t - Q^*) + \langle \nabla M_{\bar{E}}(Q_t - Q^*), Q_{t+1} - Q_t \rangle \\ &+ \tfrac{2\log(|S||A|)}{u}\|Q_{t+1} - Q_t\|_{p,\mathrm{sp}}^2. \end{aligned} \tag{4.10}$$

Next, since the span contraction assumption leads to a negative drift in (4.10) for some constant $\alpha_2$ defined in Appendix B.3,

$$\begin{aligned} \mathbb{E}\left[\langle \nabla M_{\bar{E}}(Q_t - Q^*), Q_{t+1} - Q_t\rangle\right] \le{}& M_{\bar{E}}(H^J(Q_t) - Q^*) - M_{\bar{E}}(Q_t - Q^*) \\ \le{}& -2\alpha_2 M_{\bar{E}}(Q_t - Q^*), \end{aligned} \tag{4.11}$$

we can use the smoothness and uniform approximation properties of $M_{\bar{E}}$ to provide a recursive bound of $Q_{t+1}$ for some $\alpha_3$ and $\alpha_4$ defined in Appendix B.3,

$$\mathbb{E}[M_{\bar{E}}(Q_{t+1} - Q^*)] \le (1 - 2\alpha_2\eta_t + \alpha_3\eta_t^2)M_{\bar{E}}(Q_t - Q^*) + \alpha_4\eta_t^2. \tag{4.12}$$

Clearly, by taking a small enough step size $\eta_t$, (4.12) implies the convergence of $\mathbb{E}\left[M_{\bar{E}}(Q_t - Q^*)\right]$. Now we can state a sample complexity upper bound for Algorithm 2 by choosing a specific step-size sequence.

**Theorem 2.** *If $\{Q_t\}$ is generated by Algorithm 2 with a decaying step-size sequence*

$$\eta_t = \frac{4}{1-\gamma}\frac{1}{t+K}, \text{ with } K = \max\left\{\frac{288\log(|S||A|)}{(1-\gamma)^2}, 3\right\},$$

*then for some universal constant $C$ the following bound holds for all $t \ge 1$,*

$$\mathbb{E}\left[\|Q_t - Q^*\|_{\infty,\mathrm{sp}}^2\right] \le C\left(\frac{\|Q_0 - Q^*\|_{\infty,\mathrm{sp}}^2(\log|S||A|)^2}{(1-\gamma)^4 t^2} + \frac{(J^2 + \|Q^*\|_{\infty,\mathrm{sp}}^2)\log(|S||A|)}{(1-\gamma)^3 t}\right).$$

*If instead a constant step size $\eta \le \frac{(1-\gamma)^2}{288\log(|S||A|)}$ is employed, then*

$$\mathbb{E}\left[\|Q_t - Q^*\|_{\infty,\mathrm{sp}}^2\right] \le \sqrt{e}(1 - \tfrac{1-\gamma}{2}\eta)^t\|Q_0 - Q^*\|_{\infty,\mathrm{sp}}^2 + \frac{48(J^2 + \|Q^*\|_{\infty,\mathrm{sp}}^2)\log(|S||A|)}{(1-\gamma)^2}\eta.$$

In both cases, it takes

$$\mathcal{O}\left(\frac{(J^2 + \|Q^*\|^2_{\infty,\mathrm{sp}})J|S||A|\log(|S||A|)}{(1-\gamma)^3\epsilon^2}\right)$$

samples to find a $Q_t$ with $\mathbb{E}\left[\|Q_t - Q^*\|_{\infty,\mathrm{sp}}\right] \leq \epsilon$. Taking into account $\|Q^*\|_{\infty,\mathrm{sp}} \leq \frac{J}{1-\gamma}$ from the span contraction assumption, the sample complexity can be simplified further to

$$\tilde{\mathcal{O}}\left(\frac{|S||A|J^3}{(1-\gamma)^5\epsilon^2}\right),$$

which is similar to the sample complexity of $\gamma$-discounted $Q$-learning algorithm [18, 43].

## 5 Numerical Experiments

In this section we present empirical results of the average-reward TD($\lambda$) with linear function approximation (i.e. Algorithm 1). In our simulation, we consider a randomly generated MRP with $|\mathcal{S}| = 100$ states and a randomly generated feature matrix $\Phi$ with $d = 20$ features and $e \in W_\Phi$. Experimental details and figures are provided in Appendix C. All the implementations are publicly available[③].

We first show that if the algorithm starts from different initial points, it will converge to different TD fixed points. To demonstrate that, we implement the algorithm using diminishing step sizes for 4 different $\theta_0$, and then plot $\mathbb{E}\left[\|\Pi_{2,E}(\theta_t - \theta^*)\|_2\right]$ and $\mathbb{E}\left[(\theta_t - \theta^*)^\top \frac{\theta_e}{\|\theta_e\|_2}\right]$ as functions of the number of iterations $t$ in Figure 1 and 2, respectively. Recall from Lemma 1 that $\{\theta^* + c\theta_e | c \in \mathbb{R}\}$ is the set of TD limit points. We observe in Figure 1 that $\mathbb{E}\left[\|\Pi_{2,E}(\theta_t - \theta^*)\|_2\right]$ converges to 0 for all 4 initial points, which means that the iterates $\theta_t$ converge to some TD limit point regardless of $\theta_0$. Moreover, Figure 2 shows that $\mathbb{E}\left[(\theta_t - \theta^*)^\top \frac{\theta_e}{\|\theta_e\|_2}\right]$ converges to different values for different initial points. This, combined with Figure 1, implies that the algorithm converges to different TD limit points, starting from different $\theta_0$.

We next numerically verify the finite-time error bounds of Algorithm 1 using decaying step sizes. In Figure 3, we plot $\mathbb{E}\left[(\bar{r}_t - r^*)^2 + \|\Pi_{2,E}(\theta_t - \theta^*)\|_2^2\right]$ as a function of $t$ for $\lambda \in \{0, 0.2, 0.4, 0.8\}$, where $r^*$ denotes the average-reward of the MRP. We see that the iterates $\{(\bar{r}_t, \theta_t)\}$ of the algorithm converge for all values of $\lambda$. To further verify the rate of convergence, we plot $\ln \mathbb{E}\left[(\bar{r}_t - r^*)^2 + \|\Pi_{2,E}(\theta_t - \theta^*)\|_2^2\right]$ as a function of $\ln t$ in Figure 4 for large $t$ and the slopes of these lines are provided in the legend. The plot shows $\mathbb{E}\left[(\bar{r}_t - r^*)^2 + \|\Pi_{2,E}(\theta_t - \theta^*)\|_2^2\right] \approx \mathcal{O}\left(\frac{1}{t}\right)$ asymptotically, which agrees with Theorem 1(b). In addition, we notice from Figure 3 and 4 that the best $\lambda$ in terms of sample complexity is 0.2, which confirms our Remark 2 that intermediate value of $\lambda$ yields the best performance with regard to sample complexity.

## 6 Conclusion

We establish the first finite sample convergence bounds of (i) average-reward TD($\lambda$) with linear function approximation under Markovian observation noise, and (ii) average-reward tabular $Q$-learning in the synchronous setting. These RL algorithms can be viewed as SA schemes to solve average-reward Bellman equations. However, the Bellman operators are not contractive under any norm. To resolve this difficulty, we construct Lyapunov functions using projection and infimal convolution to analyze the convergence of equivalent classes generated by these algorithms. Our approach is simple and general, so we expect it to have broader applications in other problems.

When analyzing the average-reward $Q$-learning algorithm, we made a $J$-step span contraction assumption (i.e. Assumption 4), which is not needed for the asymptotic convergence [10]. However, it is unclear if such an assumption is necessary for establishing any finite-time convergence bound. Since our results are the first finite sample bounds, a future research direction is on relaxing this assumption. Besides, it would be interesting to experiment our algorithms in practice and see how the empirical performance is compared with our theoretical bounds.

---

[③]https://github.com/xiaojianzhang/Average-Reward-TD-Q-Learning

## Acknowledgment

This work was partially supported by an award from Raytheon Technologies and a seed grant from Georgia Institute of Technology.

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
