# Appendices

## A Proofs in Section 3

### A.1 Proof of Lemma 1

**1.** When $e \notin W_\Phi$, we have $E = \mathbb{R}^d$ and $W_{\Phi,E} = W_\Phi$. By Theorem 1 in [10], we know that the projected Bellman equation (3.4) has a unique fixed point $\theta^*$. Thus, $\mathcal{L} = \{\theta^*\}$.

**2.** When $e \in W_\Phi$, $\theta_e$ is a unique solution to $\Phi\theta = e$ as $\Phi$ is full column rank. We first show that the set of solutions to the projected Bellman equation (3.4) takes the form $\{\tilde{\theta} + c\theta_e | c \in \mathbb{R}\}$, where $\tilde{\theta}$ is any solution to (3.4). Let $\theta := \tilde{\theta} + c\theta_e$ for any scalar $c$. Then,

$$
\begin{aligned}
\Pi_{D,W_\Phi} T^{(\lambda)} \Phi\theta &= \Pi_{D,W_\Phi} T^{(\lambda)} \Phi\left(\tilde{\theta} + c\theta_e\right) \\
&= \Pi_{D,W_\Phi} T^{(\lambda)} \left(\Phi\tilde{\theta} + ce\right) \\
&= \Pi_{D,W_\Phi} T^{(\lambda)} \Phi\tilde{\theta} + ce \\
&= \Phi\tilde{\theta} + c\Phi\theta_e \\
&= \Phi\left(\tilde{\theta} + c\theta_e\right) \\
&= \Phi\theta.
\end{aligned}
$$

On the other hand, suppose that $\theta$ is not of the form $\tilde{\theta} + c\theta_e$. Then,

$$
\begin{aligned}
\Pi_{D,W_\Phi} T^{(\lambda)} \Phi\theta &= \Pi_{D,W_\Phi} T^{(\lambda)} \Phi\left(\theta - \tilde{\theta} + \tilde{\theta}\right) \\
&= \Pi_{D,W_\Phi} T^{(\lambda)} \Phi\tilde{\theta} + \Pi_{D,W_\Phi} P^{(\lambda)} \Phi\left(\theta - \tilde{\theta}\right) \\
&= \Phi\tilde{\theta} + \Pi_{D,W_\Phi} P^{(\lambda)} \Phi\left(\theta - \tilde{\theta}\right) \\
&\neq \Phi\tilde{\theta} + \Phi\left(\theta - \tilde{\theta}\right) \\
&= \Phi\theta,
\end{aligned}
$$

where the "not equal to" is due to Lemma 2 in [10] and the non-expansiveness of the projection $\Pi_{D,W_\Phi}$.

As the set of solutions to Eq. (3.4) is a line parallel to the subspace $\{c\theta_e | c \in \mathbb{R}\}$ and $E$ is the orthogonal complement of $\{c\theta_e | c \in \mathbb{R}\}$, there is a unique solution of Eq. (3.4) that lies in $E$. We refer to this particular solution as $\theta^*$. It then follows that $\theta^*$ is also a solution to $\Phi\theta = \Pi_{D,W_{\Phi,E}} T^{(\lambda)} \Phi\theta$.

Now we just need to show that the solution to $\Phi\theta = \Pi_{D,W_{\Phi,E}} T^{(\lambda)} \Phi\theta$ is unique. We notice that the equation $\Phi\theta = \Pi_{D,W_{\Phi,E}} T^{(\lambda)} \Phi\theta$ is equivalent to

$$
\underbrace{\Pi_{2,E} \Phi^\top D \left(P^{(\lambda)} - I\right) \Phi}_{A'} \theta = \underbrace{\Pi_{2,E} \Phi^\top D \left[\frac{r(\mu)}{1-\lambda} e - \mathcal{R}^{(\lambda)}\right]}_{b'},
$$

where $\mathcal{R}^{(\lambda)} = (1-\lambda) \sum_{m=0}^{\infty} \lambda^m \sum_{k=0}^{m} P^k \mathcal{R}$.

Suppose $\theta^*$ is a solution of the equation $\Phi\theta = \Pi_{D,W_{\Phi,E}} T^{(\lambda)} \Phi\theta$. Then we know that $\theta^*$ must lie in the subspace $E$. Thus, we have $\Phi\theta^* = \Phi\Pi_{2,E}\theta^*$. By the definition of the projection operator $\Pi_{D,W_{\Phi,E}}$, we have

$$
\Pi_{D,W_{\Phi,E}} V = \operatorname{argmin}_{\bar{V} \in \{\Phi\theta | \theta \in E\}} \|V - \bar{V}\|_D = \operatorname{argmin}_{\bar{V} \in \{\Phi\Pi_{2,E}\theta | \theta \in \mathbb{R}^d\}} \|V - \bar{V}\|_D.
$$

Therefore, using $\Phi\theta^* = \Phi\Pi_{2,E}\theta^*$, the equation $\Phi\theta^* = \Pi_{D,W_{\Phi,E}} T^{(\lambda)} \Phi\theta^*$ is equivalent to

$$
\Phi\Pi_{2,E}\theta^* \in \operatorname{argmin}_{\bar{V} \in \{\Phi\Pi_{2,E}\theta | \theta \in \mathbb{R}^d\}} \|T^{(\lambda)} \Phi\Pi_{2,E}\theta^* - \bar{V}\|_D.
$$

Thus, by the first-order optimality condition and the definition of $T^{(\lambda)}$, we have

$$\Pi_{2,E}\Phi^\top D \left[ \mathcal{R}^{(\lambda)} - \frac{r(\mu)}{1-\lambda} e + P^{(\lambda)}\Phi\Pi_{2,E}\theta^* - \Phi\Pi_{2,E}\theta^* \right] = 0$$

Using $\Phi\theta^* = \Phi\Pi_{2,E}\theta^*$ again and rearranging terms, we have

$$\Pi_{2,E}\Phi^\top D \left( P^{(\lambda)} - I \right) \Phi\theta = \Pi_{2,E}\Phi^\top D \left[ \frac{r(\mu)}{1-\lambda}e - \mathcal{R}^{(\lambda)} \right].$$

On the other hand, suppose $\theta^*$ is in the subspace $E$ and satisfies

$$\Pi_{2,E}\Phi^\top D \left( P^{(\lambda)} - I \right) \Phi\theta = \Pi_{2,E}\Phi^\top D \left[ \frac{r(\mu)}{1-\lambda}e - \mathcal{R}^{(\lambda)} \right].$$

Then, following the same arguments above reversely, we can show that $\theta^*$ is a solution of the equation

$$\Phi\theta = \Pi_{D,W_\Phi,E} T^{(\lambda)}\Phi\theta.$$

For any $\theta \in E$, we have

$$\begin{aligned}
\theta^\top A'\theta &= \theta^\top \Pi_{2,E}\Phi^\top D \left( P^{(\lambda)} - I \right) \Phi\theta \\
&= \theta^\top \Pi_{2,E}^\top \Phi^\top D \left( P^{(\lambda)} - I \right) \Phi\theta \\
&= \left( \Pi_{2,E}\theta \right)^\top \Phi^\top D \left( P^{(\lambda)} - I \right) \Phi\theta \\
&= \theta^\top \Phi^\top D \left( P^{(\lambda)} - I \right) \Phi\theta \\
&\leq -\Delta \|\theta\|_2^2,
\end{aligned}$$

where the last inequality is due to Lemma 2. Suppose $A'\theta_1 = b'$ and $A'\theta_2 = b'$ for some $\theta_1, \theta_2 \in E$. Then, $0 = (\theta_1 - \theta_2)^\top A'(\theta_1 - \theta_2) \leq -\Delta \|\theta_1 - \theta_2\|_2^2$, which implies $\theta_1 = \theta_2$. Therefore, $\Phi\theta = \Pi_{D,W_\Phi,E} T^{(\lambda)}\Phi\theta$ has a unique solution.

## A.2 Proof of Lemma 2

For every $\theta \in E$, we have $\Phi\theta \neq e$. This is because

    (1) if $e \notin W_\Phi$, then there is no $\theta \in \mathbb{R}^d = E$ satisfying $\Phi\theta = e$.

    (2) if $e \in W_\Phi$, then $\theta_e \notin E$ is the unique solution to $\Phi\theta = e$.

Thus $V_\theta := \Phi\theta$ is a non-constant vector in $\mathbb{R}^{|\mathcal{S}|}$ for any $\theta \in E$. Using the fact proved in Lemma 7 of [10] that $J^\top D \left( I - P^{(\lambda)} \right) J > 0$ for any non-constant vector $J \in \mathbb{R}^{|\mathcal{S}|}$, for any non-zero $\theta \in E$, we have

$$\theta^\top \Phi^\top D \left( I - P^{(\lambda)} \right) \Phi\theta = V_\theta^\top D \left( I - P^{(\lambda)} \right) V_\theta > 0.$$

Since the set $\{\theta \in E | \|\theta\|_2 = 1\}$ is nonempty and compact, by the extreme value theorem, we have

$$\Delta := \min_{\|\theta\|_2=1, \theta \in E} \theta^\top \Phi^\top D \left( I - P^{(\lambda)} \right) \Phi\theta > 0.$$

Under Assumption 1, the steady-state expectations $A := \mathbb{E}_\pi \left[ A(X_t) \right]$ is given by

$$A = \begin{bmatrix} -c_\alpha & 0 \\ -\frac{1}{1-\lambda}\Phi^\top De & \Phi^\top D \left( P^{(\lambda)} - I \right) \Phi \end{bmatrix},$$

We first rewrite the minimization problem $\min_{\|\Theta\|_2=1, \Theta \in \mathbb{R} \times E} -\Theta^\top A\Theta$ as

$$\min_{\sqrt{\bar{r}^2 + \|\theta\|_2^2}=1, \bar{r} \in \mathbb{R}, \theta \in E} c_\alpha \bar{r}^2 + \frac{\bar{r}}{1-\lambda}\theta^\top \Phi^\top De + \theta^\top \Phi^\top D \left( I - P^{(\lambda)} \right) \Phi\theta.$$

Since

$$\left| \frac{\bar{r}}{1-\lambda} \theta^\top \Phi^\top D e \right| = \frac{|\bar{r}|}{1-\lambda} \left| \theta^\top \Phi^\top D e \right|$$

$$= \frac{|\bar{r}|}{1-\lambda} \left| (\Phi\theta)^\top \pi \right|$$

$$\leq \frac{|\bar{r}|}{1-\lambda} \|\pi\|_1 \|\Phi\theta\|_\infty$$

$$= \frac{|\bar{r}|}{1-\lambda} \|\Phi\theta\|_\infty$$

$$\leq \frac{|\bar{r}|}{1-\lambda} \max_{i\in\mathcal{S}} \|\phi(i)\|_2 \|\theta\|_2$$

$$\leq \frac{|\bar{r}| \|\theta\|_2}{1-\lambda}, \quad \forall \bar{r}\in\mathbb{R}, \theta\in E,$$

and

$$\theta^\top \Phi^\top D \left(I - P^{(\lambda)}\right) \Phi\theta \geq \Delta \|\theta\|_2^2, \quad \forall\theta\in E,$$

then we have

$$\min_{\sqrt{\bar{r}^2+\|\theta\|_2^2}=1, \bar{r}\in\mathbb{R}, \theta\in E} c_\alpha \bar{r}^2 + \frac{\bar{r}}{1-\lambda}\theta^\top\Phi^\top D e + \theta^\top\Phi^\top D\left(I - P^{(\lambda)}\right)\Phi\theta$$

$$\geq \min_{\sqrt{\bar{r}^2+\|\theta\|_2^2}=1, \bar{r}\in\mathbb{R}, \theta\in E} c_\alpha \bar{r}^2 - \frac{|\bar{r}|\|\theta\|_2}{1-\lambda} + \Delta\|\theta\|_2^2$$

$$= \min_{\bar{r}\in[-1,1]} c_\alpha|\bar{r}|^2 - \frac{1}{1-\lambda}|\bar{r}|\sqrt{1-|\bar{r}|^2} + \Delta\left(1-|\bar{r}|^2\right)$$

$$= \min_{x\in[0,1]} c_\alpha x - \frac{1}{1-\lambda}\sqrt{x(1-x)} + \Delta(1-x)$$

$$= \Delta + \min_{x\in[0,1]} (c_\alpha - \Delta)x - \frac{1}{1-\lambda}\sqrt{x(1-x)}.$$

When $c_\alpha \geq \Delta + \sqrt{\frac{1}{\Delta^2(1-\lambda)^4} - \frac{1}{(1-\lambda)^2}}$, we have

$$\min_{x\in[0,1]} (c_\alpha - \Delta)x - \frac{1}{1-\lambda}\sqrt{x(1-x)} \geq -\frac{\Delta}{2},$$

which implies that

$$\min_{\|\Theta\|_2=1, \Theta\in\mathbb{R}\times E} -\Theta^\top A\Theta \geq \frac{\Delta}{2}.$$

### A.3 Proof of Theorem 1

*Proof.* **Part (1): auxiliary algorithm.** Suppose the sequence of iterates $\{(\bar{r}_t, \theta_t)\}$ is generated by Algorithm 1. Then, the sequence of iterates $\{(\bar{r}_t, \Pi_{2,E}\theta_t)\}$ can be generated by the following auxiliary algorithm

$$\bar{r}_{t+1} = \bar{r}_t + c_\alpha\beta_t(\mathcal{R}(s_t) - \bar{r}_t) \text{ and } \theta_{t+1} = \theta_t + \beta_t\delta_t(\theta_t)\Pi_{2,E}z_t, \quad (A.1)$$

with initial values $\bar{r}_0$ and $\Pi_{2,E}\theta_0$. Note that the iterates $\{(\bar{r}_t, \theta_t)\}$ uniquely determines the iterates $\{(\bar{r}_t, \Pi_{2,E}\theta_t)\}$.

The auxiliary algorithm (A.1) can be rewritten in the following vector form

$$\Theta_{t+1} = \Theta_t + \beta_t \left[\tilde{A}(X_t)\Theta_t + \tilde{b}(X_t)\right], \quad (A.2)$$

where

$$\tilde{A}(X_t) = \begin{bmatrix} -c_\alpha & 0 \\ -\Pi_{2,E}z_t & \Pi_{2,E}z_t\left(\phi(s_{t+1})^\top - \phi(s_t)^\top\right) \end{bmatrix}$$

and

$$\tilde{b}(X_t) = \begin{bmatrix} c_\alpha \mathcal{R}(s_t) \\ \mathcal{R}(s_t)\Pi_{2,E}z_t \end{bmatrix}.$$

If we define

$$\Pi := \begin{bmatrix} 1 & 0 \\ 0 & \Pi_{2,E} \end{bmatrix},$$

then we have $\tilde{A}(X_t) = \Pi A(X_t)$ and $\tilde{b}(X_t) = \Pi b(X_t)$.

Under Assumption 1, the steady-state expectations $\tilde{A} := \mathbb{E}_\pi\left[\tilde{A}(X_t)\right]$ and $\tilde{b} := \mathbb{E}_\pi\left[\tilde{b}(X_t)\right]$ are given by

$$\tilde{A} = \Pi A = \begin{bmatrix} -c_\alpha & 0 \\ -\frac{1}{1-\lambda}\Pi_{2,E}\Phi^\top De & \Pi_{2,E}\Phi^\top D\left(P^{(\lambda)} - I\right)\Phi \end{bmatrix},$$

and

$$\tilde{b} = \Pi b = \begin{bmatrix} c_\alpha r(\mu) \\ \Pi_{2,E}\Phi^\top D\mathcal{R}^{(\lambda)} \end{bmatrix}.$$

Stochastic approximation theory shows that the asymptotic behavior of the sequence $\{(\bar{r}_t, \Pi_{2,E}\theta_t)\}$ generated by (A.1) is closely linked with the corresponding ordinary differential equation $\dot{\Theta}_t = \tilde{A}\Theta_t + \tilde{b}$ and the limit point of $\{(\bar{r}_t, \Pi_{2,E}\theta_t)\}$ should satisfies the equation $\tilde{A}\Theta + \tilde{b} = 0$. Solving this equation, we have the limit point of $\{(\bar{r}_t, \Pi_{2,E}\theta_t)\}$ is $(r(\mu), \theta^*)$.

We notice that

$$\begin{aligned}
&\min_{\|\Theta\|_2=1,\Theta\in\mathbb{R}\times E} -\Theta^\top\tilde{A}\Theta \\
=&\min_{\sqrt{\bar{r}^2+\|\theta\|_2^2}=1,\bar{r}\in\mathbb{R},\theta\in E} c_\alpha\bar{r}^2 + \frac{\bar{r}}{1-\lambda}\theta^\top\Pi_{2,E}\Phi^\top De + \theta^\top\Pi_{2,E}\Phi^\top D\left(I - P^{(\lambda)}\right)\Phi\theta \\
=&\min_{\sqrt{\bar{r}^2+\|\theta\|_2^2}=1,\bar{r}\in\mathbb{R},\theta\in E} c_\alpha\bar{r}^2 + \frac{\bar{r}}{1-\lambda}\theta^\top\Phi^\top De + \theta^\top\Phi^\top D\left(I - P^{(\lambda)}\right)\Phi\theta \\
=&\min_{\|\Theta\|_2=1,\Theta\in\mathbb{R}\times E} -\Theta^\top A\Theta \geq \frac{\Delta}{2}.
\end{aligned}$$

Furthermore,

$$\begin{aligned}
\left\|\tilde{A}(X_t)\right\|_2 &= \|\Pi A(X_t)\|_2 \\
&\leq \|A(X_t)\|_2 \\
&\leq \|A(X_t)\|_F \\
&= \sqrt{c_\alpha^2 + \|z_t\|_2^2 + \|z_t[\phi(s_{t+1})^\top - \phi(s_t)^\top]\|_F^2} \\
&\leq \sqrt{c_\alpha^2 + \|z_t\|_2^2 + \|z_t[\phi(s_{t+1})^\top - \phi(s_t)^\top]\|_2^2} \\
&\leq \sqrt{c_\alpha^2 + \|z_t\|_2^2 + \left(\|z_t\|_2\|\phi(s_{t+1})\|_2 + \|z_t\|_2\|\phi(s_t)\|_2\right)^2} \\
&\leq \sqrt{c_\alpha^2 + \frac{1}{(1-\lambda)^2} + \frac{4}{(1-\lambda)^2}} \\
&= \sqrt{c_\alpha^2 + \frac{5}{(1-\lambda)^2}},
\end{aligned}$$

and

$$\left\|\tilde{b}\left(X_t\right)\right\| = \|\Pi b\left(X_t\right)\|_2$$
$$\leq \|b\left(X_t\right)\|_2$$
$$\leq \sqrt{(c_\alpha \mathcal{R}(s_t))^2 + \mathcal{R}(s_t)^2\|z_t\|_2^2}$$
$$\leq \sqrt{c_\alpha^2 + \frac{1}{(1-\lambda)^2}}.$$

**Part (2): general finite-time bound.** For ease of notation, we let

$$\mathbb{E}_t\left[\cdot\right] := \mathbb{E}\left[\cdot|\Theta_{t-\tau(\beta_t)}, X_{t-\tau(\beta_t)}\right],$$

and

$$\beta_{t_1,t_2} := \sum_{k=t_1}^{t_2} \beta_k.$$

Note that in this part, $\Theta_t := \begin{bmatrix} \bar{r}_t \\ \Pi_{2,E}\theta_t \end{bmatrix}$, $\Theta^* := \begin{bmatrix} r(\mu) \\ \theta^* \end{bmatrix}$, $A(X_t) := \tilde{A}(X_t)$, $A := \tilde{A}$, $b(X_t) := \tilde{b}(X_t)$, $b := \tilde{b}$, $A_{\max} := \sqrt{c_\alpha^2 + \frac{5}{(1-\lambda)^2}}$, $b_{\max} := \sqrt{c_\alpha^2 + \frac{1}{(1-\lambda)^2}}$, $\eta := \sqrt{c_\alpha^2 + \frac{5}{(1-\lambda)^2}}$.

The step size sequence $\{\beta_t\}$ satisfies the following conditions: (i) $\{\beta_t\}$ are positive and non-increasing; (ii) there exists a smallest positive integer $t^*$ such that $\beta_{0,t^*-1} \leq \frac{1}{2\eta}$, and for all $t \geq t^*$, $\beta_{t-\tau(\beta_t),t-1} \leq \min\{\frac{1}{4\eta}, \frac{\Delta}{228\eta^2}\}$ and $\frac{\beta_{t-\tau(\beta_t),t-1}}{\tau(\beta_t)\beta_t} \leq 2$.

For ant $t \geq 0$, we have

$$\mathbb{E}_t\left[\|\Theta_{t+1} - \Theta^*\|_2^2 - \|\Theta_t - \Theta^*\|_2^2\right]$$
$$= \mathbb{E}_t\left[\|\Theta_{t+1} - \Theta_t + \Theta_t - \Theta^*\|_2^2 - \|\Theta_t - \Theta^*\|_2^2\right]$$
$$= \mathbb{E}_t\left[\|\Theta_{t+1} - \Theta_t\|_2^2 + 2\left(\Theta_t - \Theta^*\right)^\top \left(\Theta_{t+1} - \Theta_t\right)\right]$$
$$= \mathbb{E}_t\left[\|\Theta_{t+1} - \Theta_t\|_2^2\right] + 2\beta_t \mathbb{E}_t\left[\left(\Theta_t - \Theta^*\right)^\top \left(A(X_t)\Theta_t + b(X_t)\right)\right]$$
$$= \beta_t^2 \mathbb{E}_t\left[\|A(X_t)\Theta_t + b(X_t)\|_2^2\right]$$
$$+ 2\beta_t \mathbb{E}_t\left[\left(\Theta_t - \Theta^*\right)^\top \left(A(X_t)\Theta_t + b(X_t) - A\Theta_t - b\right)\right]$$
$$+ 2\beta_t \mathbb{E}_t\left[\left(\Theta_t - \Theta^*\right)^\top \left(A\Theta_t + b\right)\right]$$

**step 1.** Bounding $\|A(X_t)\Theta_t + b(X_t)\|_2^2$

Since $A(X_t)$ and $b(X_t)$ are uniformly bounded by $A_{\max}$ and $b_{\max}$ respectively, we then have

$$\|A(X_t)\Theta_t + b(X_t)\|_2 \leq \|A(X_t)\|_2\|\Theta_t\|_2 + \|b(X_t)\|_2$$
$$\leq A_{\max}\|\Theta_t\|_2 + b_{\max}$$
$$\leq \eta\left(\|\Theta_t\|_2 + 1\right),$$

which implies that

$$\|A(X_t)\Theta_t + b(X_t)\|_2^2 \leq \eta^2\left(\|\Theta_t - \Theta^* + \Theta^*\|_2 + 1\right)^2$$
$$\leq \eta^2\left(\|\Theta_t - \Theta^*\|_2 + \|\Theta^*\|_2 + 1\right)^2$$
$$\leq 2\eta^2\left[\|\Theta_t - \Theta^*\|_2^2 + \left(\|\Theta^*\|_2 + 1\right)^2\right].$$

**step 2.** Bounding $\left(\Theta_t - \Theta^*\right)^\top \left(A\Theta_t + b\right)$

Since $A\Theta^* + b = 0$ and $\min_{\|\Theta\|_2=1, \Theta \in \mathbb{R} \times E} -\Theta^\top A \Theta \geq \frac{\Delta}{2}$

$$
\begin{aligned}
(\Theta_t - \Theta^*)^\top (A\Theta_t + b) &= (\Theta_t - \Theta^*)^\top (A\Theta_t - A\Theta^*) \\
&= (\Theta_t - \Theta^*)^\top A (\Theta_t - \Theta^*) \\
&\leq -\frac{\Delta}{2} \|\Theta_t - \Theta^*\|_2^2
\end{aligned}
$$

**step 3.** Bounding $\mathbb{E}_t \left[ (\Theta_t - \Theta^*)^\top (A(X_t)\Theta_t + b(X_t) - A\Theta_t - b) \right]$

$$
\begin{aligned}
&\mathbb{E}_t \left[ (\Theta_t - \Theta^*)^\top (A(X_t)\Theta_t + b(X_t) - A\Theta_t - b) \right] \\
&= \mathbb{E}_t \left[ (\Theta_t - \Theta_{t-\tau(\beta_t)} + \Theta_{t-\tau(\beta_t)} - \Theta^*)^\top (A(X_t)\Theta_t + b(X_t) - A\Theta_t - b) \right] \\
&= \underbrace{\mathbb{E}_t \left[ (\Theta_t - \Theta_{t-\tau(\beta_t)})^\top (A(X_t)\Theta_t + b(X_t) - A\Theta_t - b) \right]}_{(A_1)} \\
&\quad + \underbrace{\mathbb{E}_t \left[ (\Theta_{t-\tau(\beta_t)} - \Theta^*)^\top (A(X_t)\Theta_t + b(X_t) - A\Theta_t - b) \right]}_{(A_2)}
\end{aligned}
$$

$$
\begin{aligned}
(A_1) &\leq \mathbb{E}_t \left[ | (\Theta_t - \Theta_{t-\tau(\beta_t)})^\top (A(X_t)\Theta_t + b(X_t) - A\Theta_t - b) | \right] \\
&\leq \mathbb{E}_t \left[ \| (\Theta_t - \Theta_{t-\tau(\beta_t)}) \|_2 \|A(X_t)\Theta_t + b(X_t) - A\Theta_t - b\|_2 \right] \\
&\leq 2\eta \mathbb{E}_t \left[ (\|\Theta_t\|_2 + 1) \|\Theta_t - \Theta_{t-\tau(\beta_t)}\|_2 \right] \\
&\leq 8\eta^2 \beta_{t-\tau(\beta_t),t-1} \mathbb{E}_t \left[ (\|\Theta_t\|_2 + 1)^2 \right] \\
&\leq 8\eta^2 \beta_{t-\tau(\beta_t),t-1} \mathbb{E}_t \left[ (\|\Theta_t - \Theta^*\|_2 + \|\Theta^*\|_2 + 1)^2 \right] \\
&\leq 16\eta^2 \beta_{t-\tau(\beta_t),t-1} \mathbb{E}_t \left[ \|\Theta_t - \Theta^*\|_2^2 + (\|\Theta^*\|_2 + 1)^2 \right]
\end{aligned}
$$

The 4th inequality holds because for any $0 \leq t_1 < t_2$ satisfying $\beta_{t_1,t_2-1} \leq \frac{1}{4\eta}$, the following inequality (see lemma 2.3 in [16] for a proof) hold:

$$
\|\Theta_{t_2} - \Theta_{t_1}\|_2 \leq 4\eta \beta_{t_1,t_2-1} (\|\Theta_{t_2}\|_2 + 1).
$$

Since we have assumed that $\beta_{t-\tau(\beta_t),t-1} \leq \frac{1}{4\eta}$, then we have

$$
2\eta \mathbb{E}_t \left[ (\|\Theta_t\|_2 + 1) \|\Theta_t - \Theta_{t-\tau(\beta_t)}\|_2 \right] \leq 8\eta^2 \beta_{t-\tau(\beta_t),t-1} \mathbb{E}_t \left[ (\|\Theta_t\|_2 + 1)^2 \right].
$$

Note that

$$
\begin{aligned}
&A(X_t)\Theta_t + b(X_t) - A\Theta_t - b \\
&= A(X_t)\Theta_{t-\tau(\beta_t)} - A\Theta_{t-\tau(\beta_t)} + b(X_t) - b \\
&\quad + A(X_t)\Theta_t - A\Theta_t - A(X_t)\Theta_{t-\tau(\beta_t)} + A\Theta_{t-\tau(\beta_t)} \\
&= \left[ (A(X_t) - A) \Theta_{t-\tau(\beta_t)} + b(X_t) - b \right] + (A(X_t) - A) (\Theta_t - \Theta_{t-\tau(\beta_t)})
\end{aligned}
$$

$$
\begin{aligned}
(A_2) &= \mathbb{E}_t \left[ (\Theta_{t-\tau(\beta_t)} - \Theta^*)^\top \left\{ \left[ (A(X_t) - A) \Theta_{t-\tau(\beta_t)} + b(X_t) - b \right] + (A(X_t) - A) (\Theta_t - \Theta_{t-\tau(\beta_t)}) \right\} \right] \\
&\leq \underbrace{| (\Theta_{t-\tau(\beta_t)} - \Theta^*)^\top \mathbb{E}_t \left[ (A(X_t) - A) \Theta_{t-\tau(\beta_t)} + b(X_t) - b \right] |}_{(A_{2,1})} \\
&\quad + \underbrace{| (\Theta_{t-\tau(\beta_t)} - \Theta^*)^\top \mathbb{E}_t \left[ (A(X_t) - A) (\Theta_t - \Theta_{t-\tau(\beta_t)}) \right] |}_{(A_{2,2})}
\end{aligned}
$$

Since
$$(A_{2,1}) \le \|\Theta_{t-\tau(\beta_t)} - \Theta^*\|_2 \left( \|\mathbb{E}_t [A(X_t)] - A\|_2 \|\Theta_{t-\tau(\beta_t)}\|_2 + \|\mathbb{E}_t [b(X_t)] - b\|_2 \right)$$
$$\le \beta_t \mathbb{E}_t \left[ \|\Theta_{t-\tau(\beta_t)} - \Theta^*\|_2 \left( \|\Theta_{t-\tau(\beta_t)}\|_2 + 1 \right) \right]$$
$$= \beta_t \mathbb{E}_t \left[ \|\Theta_{t-\tau(\beta_t)} - \Theta_t + \Theta_t - \Theta^*\|_2 \left( \|\Theta_{t-\tau(\beta_t)} - \Theta_t + \Theta_t - \Theta^* + \Theta^*\|_2 + 1 \right) \right]$$
$$\le \beta_t \mathbb{E}_t \left[ \left( \|\Theta_t - \Theta_{t-\tau(\beta_t)}\|_2 + \|\Theta_t - \Theta^*\|_2 \right) \left( \|\Theta_t - \Theta_{t-\tau(\beta_t)}\|_2 + \|\Theta_t - \Theta^*\|_2 + \|\Theta^*\|_2 + 1 \right) \right]$$
$$\le \beta_t \mathbb{E}_t \left[ \left( \|\Theta_t\|_2 + \|\Theta_t - \Theta^*\|_2 + 1 \right) \left( \|\Theta_t\|_2 + \|\Theta_t - \Theta^*\|_2 + \|\Theta^*\|_2 + 2 \right) \right]$$
$$\le \beta_t \mathbb{E}_t \left[ \left( \|\Theta^*\|_2 + 2\|\Theta_t - \Theta^*\|_2 + 1 \right) \left( 2\|\Theta_t - \Theta^*\|_2 + 2\|\Theta^*\|_2 + 2 \right) \right]$$
$$\le 4\beta_t \mathbb{E}_t \left[ \left( \|\Theta_t - \Theta^*\|_2 + \|\Theta^*\|_2 + 1 \right)^2 \right]$$
$$\le 8\beta_t \mathbb{E}_t \left[ \|\Theta_t - \Theta^*\|_2^2 + \left( \|\Theta^*\|_2 + 1 \right)^2 \right]$$
$$\le 8\eta^2 \beta_{t-\tau(\beta_t),t-1} \mathbb{E}_t \left[ \|\Theta_t - \Theta^*\|_2^2 + \left( \|\Theta^*\|_2 + 1 \right)^2 \right]$$

The 4th inequality holds because for any $0 \le t_1 < t_2$ satisfying $\beta_{t_1,t_2-1} \le \frac{1}{4\eta}$, the following inequality (see lemma 2.3 in [16] for a proof) hold:
$$\|\Theta_{t_2} - \Theta_{t_1}\|_2 \le \|\Theta_{t_2}\|_2 + 1.$$

Since we have assumed that $\beta_{t-\tau(\beta_t),t-1} \le \frac{1}{4\eta}$, then we have $\|\Theta_t - \Theta_{t-\tau(\beta_t)}\|_2 \le \|\Theta_t\|_2 + 1$. Thus, $\beta_t \mathbb{E}_t \left[ \left( \|\Theta_t - \Theta_{t-\tau(\beta_t)}\|_2 + \|\Theta_t - \Theta^*\|_2 \right) \left( \|\Theta_t - \Theta_{t-\tau(\beta_t)}\|_2 + \|\Theta_t - \Theta^*\|_2 + \|\Theta^*\|_2 + 1 \right) \right] \le \beta_t \mathbb{E}_t \left[ \left( \|\Theta_t\|_2 + \|\Theta_t - \Theta^*\|_2 + 1 \right) \left( \|\Theta_t\|_2 + \|\Theta_t - \Theta^*\|_2 + \|\Theta^*\|_2 + 2 \right) \right]$.

$$(A_{2,2}) \le 2\eta \mathbb{E}_t \left[ \|\Theta_{t-\tau(\beta_t)} - \Theta^*\|_2 \|\Theta_t - \Theta_{t-\tau(\beta_t)}\|_2 \right]$$
$$\le 8\eta^2 \beta_{t-\tau(\beta_t),t-1} \mathbb{E}_t \left[ \|\Theta_{t-\tau(\beta_t)} - \Theta^*\|_2 \left( \|\Theta_t\|_2 + 1 \right) \right]$$
$$\le 8\eta^2 \beta_{t-\tau(\beta_t),t-1} \mathbb{E}_t \left[ \left( \|\Theta_t - \Theta_{t-\tau(\beta_t)}\|_2 + \|\Theta_t - \Theta^*\|_2 \right) \left( \|\Theta_t\|_2 + 1 \right) \right]$$
$$\le 8\eta^2 \beta_{t-\tau(\beta_t),t-1} \mathbb{E}_t \left[ \left( \|\Theta_t - \Theta_{t-\tau(\beta_t)}\|_2 + \|\Theta_t - \Theta^*\|_2 \right) \left( \|\Theta_t\|_2 + 1 \right) \right]$$
$$\le 8\eta^2 \beta_{t-\tau(\beta_t),t-1} \mathbb{E}_t \left[ \left( \|\Theta_t\|_2 + \|\Theta_t - \Theta^*\|_2 + 1 \right) \left( \|\Theta_t\|_2 + 1 \right) \right]$$
$$\le 8\eta^2 \beta_{t-\tau(\beta_t),t-1} \mathbb{E}_t \left[ \left( \|\Theta^*\|_2 + 2\|\Theta_t - \Theta^*\|_2 + 1 \right) \left( \|\Theta^*\|_2 + \|\Theta_t - \Theta^*\|_2 + 1 \right) \right]$$
$$\le 16\eta^2 \beta_{t-\tau(\beta_t),t-1} \mathbb{E}_t \left[ \left( \|\Theta_t - \Theta^*\|_2 + \|\Theta^*\|_2 + 1 \right)^2 \right]$$
$$\le 32\eta^2 \beta_{t-\tau(\beta_t),t-1} \mathbb{E}_t \left[ \|\Theta_t - \Theta^*\|_2^2 + \left( \|\Theta^*\|_2 + 1 \right)^2 \right],$$

then we have
$$(A_2) = (A_{2,1}) + (A_{2,2})$$
$$\le 40\eta^2 \beta_{t-\tau(\beta_t),t-1} \mathbb{E}_t \left[ \|\Theta_t - \Theta^*\|_2^2 + \left( \|\Theta^*\|_2 + 1 \right)^2 \right].$$

Finally,
$$(A) = (A_1) + (A_2)$$
$$\le 56\eta^2 \beta_{t-\tau(\beta_t),t-1} \mathbb{E}_t \left[ \|\Theta_t - \Theta^*\|_2^2 + \left( \|\Theta^*\|_2 + 1 \right)^2 \right]$$

**step 4.** Putting together.
$$\mathbb{E}_t \left[ \|\Theta_{t+1} - \Theta^*\|_2^2 - \|\Theta_t - \Theta^*\|_2^2 \right]$$
$$\le 2\eta^2 \beta_t \beta_{t-\tau(\beta_t),t-1} \mathbb{E}_t \left[ \|\Theta_t - \Theta^*\|_2^2 + \left( \|\Theta^*\|_2 + 1 \right)^2 \right]$$
$$+ 112\eta^2 \beta_t \beta_{t-\tau(\beta_t),t-1} \mathbb{E}_t \left[ \|\Theta_t - \Theta^*\|_2^2 + \left( \|\Theta^*\|_2 + 1 \right)^2 \right]$$
$$- \Delta\beta_t \mathbb{E}_t \left[ \|\Theta_t - \Theta^*\|_2^2 \right]$$
$$\le \left( 114\eta^2 \beta_t \beta_{t-\tau(\beta_t),t-1} - \Delta\beta_t \right) \mathbb{E}_t \left[ \|\Theta_t - \Theta^*\|_2^2 \right] + 114\eta^2 \left( \|\Theta^*\|_2 + 1 \right)^2 \beta_t \beta_{t-\tau(\beta_t),t-1}$$

Hence, for any $t \geq t^*$, we have

$$\mathbb{E}\left[\|\Theta_{t+1} - \Theta^*\|_2^2\right] \leq \left(1 + 114\eta^2\beta_t\beta_{t-\tau(\beta_t),t-1} - \Delta\beta_t\right)\mathbb{E}\left[\|\Theta_t - \Theta^*\|_2^2\right]$$
$$+ 114\eta^2\left(\|\Theta^*\|_2 + 1\right)^2\beta_t\beta_{t-\tau(\beta_t),t-1}.$$

Since for any $t \geq t^*$ we have assumed

$$\beta_{t-\tau(\beta_t),t-1} \leq \frac{\Delta}{228\eta^2}, \text{ i.e., } 228\eta^2\beta_t\beta_{t-\tau(\beta_t),t-1} - \Delta\beta_t \leq 0,$$

and

$$\frac{\beta_{t-\tau(\beta_t),t-1}}{\tau(\beta_t)\beta_t} \leq 2, \text{ i.e., } \beta_t\beta_{t-\tau(\beta_t),t-1} \leq 2\tau(\beta_t)\beta_t^2$$

then

$$\mathbb{E}\left[\|\Theta_{t+1} - \Theta^*\|_2^2\right] \leq \left(1 - \frac{\Delta}{2}\beta_t\right)\mathbb{E}\left[\|\Theta_t - \Theta^*\|_2^2\right] + \frac{\xi_2}{2}\tau(\beta_t)\beta_t^2$$

Recursively using the preceding inequality, we have for all $T \geq t^*$

$$\mathbb{E}\left[\|\Theta_T - \Theta^*\|_2^2\right] \leq \mathbb{E}\left[\|\Theta_{t^*} - \Theta^*\|_2^2\right]\prod_{t=t^*}^{T-1}\left(1 - \frac{\Delta}{2}\beta_t\right) + \frac{\xi_2}{2}\sum_{t=t^*}^{T-1}\tau(\beta_t)\beta_t^2\prod_{j=t+1}^{T-1}\left(1 - \frac{\Delta}{2}\beta_j\right).$$

Since we have assumed that $\beta_{0,t^*-1} \leq \frac{1}{2\eta}$, then we have

$$\mathbb{E}\left[\|\Theta_{t^*} - \Theta^*\|_2^2\right] \leq \mathbb{E}\left[\left(\|\Theta_{t^*} - \Theta_0\|_2 + \|\Theta_0 - \Theta^*\|_2\right)^2\right]$$
$$\leq \left(\|\Theta_0\|_2 + \|\Theta_0 - \Theta^*\|_2 + 1\right)^2$$
$$= \xi_1,$$

which gives the desired finite-time bound:

$$\mathbb{E}\left[\|\Theta_T - \Theta^*\|_2^2\right] \leq \xi_1\prod_{t=t^*}^{T-1}\left(1 - \frac{\Delta}{2}\beta_t\right) + \frac{\xi_2}{2}\sum_{t=t^*}^{T-1}\tau(\beta_t)\beta_t^2\prod_{j=t+1}^{T-1}\left(1 - \frac{\Delta}{2}\beta_j\right).$$

**Part (3): Theorem 1(a).** Since

$$\prod_{t=\tau(\beta)}^{T-1}\left(1 - \frac{\Delta}{2}\beta\right) = \left(1 - \frac{\Delta}{2}\beta\right)^{T-\tau(\beta)},$$

and

$$\sum_{t=\tau(\beta)}^{T-1}\tau(\beta)\beta^2\prod_{j=t+1}^{T-1}\left(1 - \frac{\Delta}{2}\beta\right) = \beta^2\tau(\beta)\sum_{j=\tau(\beta)}^{T-1}\left(1 - \frac{\Delta}{2}\beta\right)^{T-j-1}$$
$$\leq \beta^2\tau(\beta)\sum_{j=0}^{\infty}\left(1 - \frac{\Delta}{2}\beta\right)^j$$
$$\leq \frac{\beta\tau(\beta)}{\frac{\Delta}{2}},$$

then we have for all $T \geq \tau(\beta)$

$$\mathbb{E}\left[(\bar{r}_T - r(\mu))^2\right] + \mathbb{E}\left[\|\Pi_{2,E}\left(\theta_T - \theta^*\right)\|_2^2\right]$$
$$= \mathbb{E}\left[\|\Theta_T - \Theta^*\|_2^2\right]$$
$$\leq \xi_1\left(1 - \frac{\Delta}{2}\beta\right)^{T-\tau(\beta)} + \xi_2\frac{\beta\tau(\beta)}{\Delta}.$$

**Part (4): Theorem 1(b).** We first bound the term $\prod_{t=t^*}^{T-1}\left(1-\frac{\Delta}{2}\beta_t\right)$.

$$\prod_{t=t^*}^{T-1}\left(1-\frac{\Delta}{2}\beta_t\right) = \prod_{t=t^*}^{T-1}\left(1-\frac{\Delta}{2}\frac{c_1}{t+c_2}\right)$$

$$\leq \prod_{t=t^*}^{T-1} e^{-\frac{\Delta}{2}\frac{c_1}{t+c_2}}$$

$$= e^{-\frac{\Delta}{2}c_1\sum_{t=t^*}^{T-1}\frac{1}{t+c_2}}$$

Since

$$\sum_{t=t^*}^{T-1}\frac{1}{t+c_2} \geq \int_{t^*}^{T}\frac{1}{x+c_2}dx$$

$$= \ln\left(\frac{T+c_2}{t^*+c_2}\right),$$

then we have

$$\prod_{t=t^*}^{T-1}\left(1-\frac{\Delta}{2}\beta_t\right) \leq e^{-\frac{\Delta}{2}c_1\ln\left(\frac{T+c_2}{t^*+c_2}\right)}$$

$$= \left(\frac{t^*+c_2}{T+c_2}\right)^{\frac{\Delta}{2}c_1}.$$

Next we bound the term $\sum_{t=t^*}^{T-1}\tau(\beta_t)\beta_t^2\prod_{j=t+1}^{T-1}\left(1-\frac{\Delta}{2}\beta_j\right)$.

Since $\tau_{\beta_t} \leq \tau_{\beta_T} \leq K\ln(\frac{1}{\beta_T}) = K\left[\ln(T+c_2)-\ln(c_1)\right]$ for all $t^* \leq t \leq T-1$, we have

$$\sum_{t=t^*}^{T-1}\tau(\beta_t)\beta_t^2\prod_{j=t+1}^{T-1}\left(1-\frac{\Delta}{2}\beta_j\right) \leq K\left[\ln(T+c_2)-\ln(c_1)\right]\sum_{t=t^*}^{T-1}\beta_t^2\prod_{j=t+1}^{T-1}\left(1-\frac{\Delta}{2}\beta_j\right).$$

Moreover,

$$\prod_{j=t+1}^{T-1}\left(1-\frac{\Delta}{2}\frac{c_1}{j+c_2}\right) \leq e^{-\frac{\Delta}{2}c_1\sum_{j=t+1}^{T-1}\frac{1}{t+c_2}}$$

$$\leq \left(\frac{t+c_2+1}{T+c_2}\right)^{\frac{\Delta}{2}c_1}$$

Then,

$$\sum_{t=t^*}^{T-1}\beta_t^2\prod_{j=t+1}^{T-1}\left(1-\frac{\Delta}{2}\beta_j\right) = \sum_{t=t^*}^{T-1}\frac{c_1^2}{(t+c_2)^2}\prod_{j=t+1}^{T-1}\left(1-\frac{\Delta}{2}\frac{c_1}{j+c_2}\right)$$

$$\leq \sum_{t=t^*}^{T-1}\frac{c_1^2}{(t+c_2)^2}\left(\frac{t+c_2+1}{T+c_2}\right)^{\frac{\Delta}{2}c_1}$$

$$= \frac{c_1^2}{(T+c_2)^{\frac{\Delta}{2}c_1}}\sum_{t=t^*}^{T-1}\left(\frac{t+c_2+1}{t+c_2}\right)^2(t+c_2+1)^{\frac{\Delta}{2}c_1-2}$$

$$\leq \frac{4c_1^2}{(T+c_2)^{\frac{\Delta}{2}c_1}}\sum_{t=t^*}^{T-1}(t+c_2+1)^{\frac{\Delta}{2}c_1-2}$$

When $\frac{\Delta}{2}c_1 > 1$, we have

$$\sum_{t=t^*}^{T-1}(t + c_2 + 1)^{\frac{\Delta}{2}c_1 - 2} \leq \int_0^T (x + c_2 + 1)^{\frac{\Delta}{2}c_1 - 2}\,dx$$

$$= \frac{1}{\frac{\Delta}{2}c_1 - 1}\left[(T + c_2 + 1)^{\frac{\Delta}{2}c_1 - 1} - (c_2 + 1)^{\frac{\Delta}{2}c_1 - 1}\right]$$

$$\leq \frac{1}{\frac{\Delta}{2}c_1 - 1}(T + c_2 + 1)^{\frac{\Delta}{2}c_1 - 1}$$

Therefore,

$$\sum_{t=t^*}^{T-1}\beta_t^2\prod_{j=t+1}^{T-1}\left(1 - \frac{\Delta}{2}\beta_j\right) \leq \frac{4c_1^2}{\frac{\Delta}{2}c_1 - 1}\frac{1}{T + c_2 + 1}\left(\frac{T + c_2 + 1}{T + c_2}\right)^{\frac{\Delta}{2}c_1}$$

$$\leq \frac{4c_1^2}{\frac{\Delta}{2}c_1 - 1}\frac{1}{T + c_2 + 1}e^{\frac{\frac{\Delta}{2}c_1}{T + c_2}}$$

$$= \frac{4c_1^2}{\frac{\Delta}{2}c_1 - 1}\frac{1}{T + c_2 + 1}e^{\frac{\Delta}{2}\beta_T}$$

Since

$$\frac{\Delta}{2}\beta_T \leq \frac{\Delta}{2}\beta_0 < 1,$$

we have

$$\sum_{t=t^*}^{T-1}\beta_t^2\prod_{j=t+1}^{T-1}\left(1 - \frac{\Delta}{2}\beta_j\right) \leq \frac{4ec_1^2}{\frac{\Delta}{2}c_1 - 1}\frac{1}{T + c_2 + 1}$$

Hence,

$$\mathbb{E}\left[(\bar{r}_T - r(\mu))^2\right] + \mathbb{E}\left[\|\Pi_{2,E}(\theta_T - \theta^*)\|_2^2\right]$$

$$= \mathbb{E}\left[\|\Theta_T - \Theta^*\|_2^2\right]$$

$$\leq \xi_1\left(\frac{t^* + c_2}{T + c_2}\right)^{\frac{\Delta}{2}c_1} + \xi_2\frac{8ec_1^2 K}{\Delta c_1 - 2}\frac{\ln(T + c_2) - \ln(c_1)}{T + c_2 + 1}$$

$\square$

## A.4 Proof of Corollary 1

*Proof.* For the diminishing step-size $\beta_t = \frac{c_1}{t + c_2}$, we choose $c_1 := \frac{4}{\Delta}, c_2 := 4$ and $c_\alpha := \Delta + \frac{1}{\Delta(1-\lambda)^2}$. Then, from Theorem 1 (b), we have

$$\mathbb{E}\left[(\bar{r}_T - r(\mu))^2\right] + \mathbb{E}\left[\|\Pi_{2,E}(\theta_T - \theta^*)\|_2^2\right] \leq \xi_1\left(\frac{t^* + 4}{T + 4}\right)^2 + \frac{64eK\xi_2}{\Delta^2}\frac{\ln(T + 4) - \ln(\frac{4}{\Delta})}{T + 5}.$$

For any $\epsilon > 0$, to guarantee that $\mathbb{E}\left[|\bar{r}_T - r(\mu)|\right] \leq \epsilon$ and $\mathbb{E}\left[\|\Pi_{2,E}(\theta_T - \theta^*)\|_2\right] \leq \epsilon$, we can set

$$\xi_1\left(\frac{t^* + 4}{T + 4}\right)^2 \leq \frac{1}{2}\epsilon^2,$$

and

$$\frac{64eK\xi_2}{\Delta^2}\frac{\ln(T + 4) - \ln(\frac{4}{\Delta})}{T + 5} \leq \frac{1}{2}.\epsilon^2$$

Then $T$ needs to satisfy

$$T = \tilde{\mathcal{O}}\left(\frac{K\log\left(\frac{1}{\Delta}\right)\left(1 + \|\theta^*\|_2^2\right)}{\Delta^4(1-\lambda)^4\epsilon^2}\right)$$

$\square$

# B Convergence with respect to Span Lyapunov Function

The main difficulty for analyzing average reward is the existence of some subspace $\bar{E}$ for which the Bellman operator $H$ is indifferent, i.e.,

$$H(Q + x) - H(Q) \in \bar{E}, \quad \forall x \in \bar{E}$$

So it is impossible to apply the finite time analysis in the literature to establish the convergence of the iterates to some fix point. In essence, $H$ operates on sets of points defined by the indifferent subspace called equivalent classes:

$$\mathcal{X}_{\bar{E}} := \{x_{\bar{E}} | x \in \mathbb{R}^n\},$$

where $x_{\bar{E}} := \{y \in \mathbb{R}^n : y - x \in \bar{E}\}$. Thus we should analyze those equivalent classes rather than the points. Towards that end, we propose a new kind of Lyapunov function defined with respect to $\mathcal{X}_{\bar{E}}$.

## B.1 The Semi-Lyapunov Function

We tweak the smooth convex Lyapunov function $M$ introduced in [18] to build a new Lyapunov function. Recall that $M$ satisfies the following two important properties with respect to a smoothness norm $\|\cdot\|_s$ and a contraction norm $\|\cdot\|_c$:

1. Smoothness: $M(y) \le M(x) + \langle \nabla M(x), y - x \rangle + \frac{L}{2} \|y - x\|_s^2, \forall x, y$ for some $L \ge 0$.

2. Uniform Approximation: For some constants $c_l, c_u \ge 0$, we have

$$c_l M(x) \le \frac{1}{2} \|x\|_c^2 \le c_u M(x) \, \forall x \tag{B.1}$$

Next we construct a Lyapunov function satisfying the above two properties with respect to equivalent classes. Consider the fellowing span norm induced by $\bar{E}$ [44]:

$$\|x\|_{c,\bar{E}} := \inf_{e \in \bar{E}} \|x - e\|_c, \|x\|_{s,\bar{E}} := \inf_{e \in \bar{E}} \|x - e\|_s.$$

Clearly they are functions defined on $\mathcal{X}_{\bar{E}}$ since any element of an equivalent class $x_{\bar{E}}$ is mapped to the same value.

A key observation is that they could be expressed equivalently as the infimal convolution with respect to indicator functions. More specifically, if $\delta_{\bar{E}}$ denote the indicator function with respect $\bar{E}$,

$$\delta_{\bar{E}}(x) := \begin{cases} 0 & x \in \bar{E}, \\ \infty & \text{otherwise.} \end{cases} \tag{B.2}$$

Then $\|x\|_{c,\bar{E}} \equiv (\|\cdot\|_c \,\square\, \delta_{\bar{E}})(x)$, $\|x\|_{s,\bar{E}} \equiv (\|\cdot\|_s \,\square\, \delta_{\bar{E}})(x)$. Indeed, our new Lyapunov function $M_{\bar{E}}$ is defined as

$$M_{\bar{E}}(x) := \inf_y M(x - y) + \delta_{\bar{E}}(y) \equiv M \square \delta_{\bar{E}}(x). \tag{B.3}$$

We call it a *semi-Lyapunov function* because $M_{\bar{E}}(x) = 0 \,\forall\, x \in \bar{E}$. Notice that function $M_{\bar{E}}$ is a well-defined over $\mathcal{X}_{\bar{E}}$.

Now we show that $M_{\bar{E}}$ is a uniform approximation to the induced contraction norm $\|\cdot\|_{c,\bar{E}}$ and that it is smooth with respect to the induced smoothness norm $\|\cdot\|_{s,\bar{E}}$. First, the following properties for infimal convolution of an indicator function can be derived easily from the definition of infimal convolution.

**Lemma 4.** *Let $\bar{E}$ be a linear subspace in $\mathbb{R}^n$ and let $\delta_{\bar{E}}$ be the indicator function associated with it* (B.2). *Then the following properties hold.*

   a) *Monotonicity: If $f(x) \ge g(x)$, then $f\square\delta_{\bar{E}}(x) \ge g\square\delta_{\bar{E}}(x)$.*

   b) *Scaling Invariance: $(\beta f)\square\delta_{\bar{E}} \equiv \beta(f\square\delta_{\bar{E}})$ for any non-negative scalar $\beta$.*

   c) *Commutativity $f\square g \equiv g\square f$.*

   d) *Associativity: $(f\square g)\square h \equiv f\square(g\square h)$.*

*e)* $\delta_{\bar{E}} \Box \delta_{\bar{E}} \equiv \delta_{\bar{E}}$.

*f) If $f$ is L-smooth with respect to $\|\cdot\|_s$, then $f \Box \delta_{\bar{E}}$ is also smooth with respect to $\|\cdot\|_s$.*

*g) If $f$ is convex, then $f \Box \delta_{\bar{E}}$ is also convex.*

*Proof.* Properties f) and g) are the smoothing properties of infimal convolution and their derivations can be found in [45]. □

Then the uniform approximation property of $M_{\bar{E}}$ follows from that of $M$.

**Proposition 1.** *If $M$ satisfies $c_l M(x) \le \frac{1}{2} \|x\|_c^2 \le c_u M(x)$ for some constants $c_l, c_u$, then $M_{\bar{E}}$ defined in* (B.3) *satisfies*

$$c_l M_{\bar{E}}(x) \le \tfrac{1}{2} \|x\|_{c,\bar{E}}^2 \le c_u M_{\bar{E}}(x), \quad \forall x. \tag{B.4}$$

*Proof.* By the monotonicity of square for positive scalar, we have

$$\|x\|_{c,\bar{E}}^2 = (\inf_y \|x - y\|_c + \delta_{\bar{E}}(y))^2 = \inf_y(\|x - y\|_c + \delta_{\bar{E}}(y))^2 \overset{(a)}{=} \inf_y \|x - y\|_c^2 + \delta_{\bar{E}}(y) = \|\cdot\|_c^2 \Box \delta_{\bar{E}},$$

where (a) follows from $\delta_{\bar{E}}$ being a support function. The monotonicity of the infimal convolution Lemma 4.a) implies that

$$(c_l M) \Box \delta_{\bar{E}}(x) \le \tfrac{1}{2} \|\cdot\|_c^2 \Box \delta_{\bar{E}}(x) \le (c_u M) \Box \delta_{\bar{E}}(x), \quad \forall x.$$

So the Lemma 4.b) implies

$$c_l(M \Box \delta_{\bar{E}})(x) \le \tfrac{1}{2} \|\cdot\|_{c,\bar{E}}^2(x) \le c_u(M \Box \delta_{\bar{E}})(x), \quad \forall x,$$

i.e.,

$$c_l M_{\bar{E}}(x) \le \tfrac{1}{2} \|x\|_{c,\bar{E}}^2 \le c_u M_{\bar{E}}(x), \quad \forall x.$$

□

Moreover the smoothness of $M_{\bar{E}}$ also follows from that of $M$.

**Proposition 2.** *If $M$ is L-smooth with respect to $\|\cdot\|_s$,*

$$M(y) \le M(x) + \langle \nabla M(x), y - x \rangle + \tfrac{L}{2} \|y - x\|_s^2, \forall x, y,$$

*then $M_{\bar{E}}$ is L-smooth with respect to $\|\cdot\|_{s,\bar{E}}$, i.e.,i.e,*

$$M_{\bar{E}}(y) \le M_{\bar{E}}(x) + \langle \nabla M_{\bar{E}}(x), y - x \rangle + \tfrac{L}{2} \|y - x\|_{s,\bar{E}}^2, \forall x, y.$$

*Moreover, the gradient of $M_{\bar{E}}$ satisfies $\langle \nabla M_{\bar{E}}(x), e \rangle = 0 \ \forall e \in E, \forall x$.*

*Proof.* $\langle \nabla M_{\bar{E}}(x), e \rangle = 0, \forall e \in E$ clearly holds because $M_{\bar{E}}$ always have the same value for any elements of $x_{\bar{E}}$. Now we show the smoothness property. First, by Lemma 4.f), if $M$ is $L$-smooth with respect to $\|\cdot\|_s$, then $M_{\bar{E}}$ must also be $L$-smooth with respect to $\|\cdot\|_s$. Now consider arbitrary $x, y \in \mathbb{R}^n$. Let $\hat{e} = \arg\min_{e \in \bar{E}} \|x - y - e\|_s$, i.e., $\|x - y - \hat{e}\|_s = \|x - y\|_{s,\bar{E}}$. Then

$$M_{\bar{E}}(x) = M_{\bar{E}}(x + \hat{e}) \overset{(a)}{\le} M_{\bar{E}}(y) + \langle \nabla M_{\bar{E}}(y), x + \hat{e} - y \rangle + \tfrac{L}{2} \|x + \hat{e} - y\|_s^2$$
$$= M_{\bar{E}}(y) + \langle \nabla M_{\bar{E}}(y), x - y \rangle + \tfrac{L}{2} \|x - y\|_{s,\bar{E}}^2,$$

where (a) follows from the $L$-smoothness of $M_{\bar{E}}$ with respect to $\|\cdot\|_s$. □

## B.2 Recursive Bounds of the General Stochastic Approximation Scheme

Now let's analyze the iterates generated by the following stochastic approximation scheme for solving some fixed equivalent class equation $H(x) - x \in \bar{E}$:

$$x^{t+1} \leftarrow x^t + \eta_t(\hat{H}(x^t) - x^t), \tag{B.5}$$

We make the following assumptions regarding the function $H$ and its stochastic sample $\hat{H}$.

**Assumption 4.**

1. $H$ is $\gamma$-contractive with respective to $\|\cdot\|_{c,\bar{E}}$ for some $\gamma < 1$, i.e., $\|H(x) - H(y)\|_{c,\bar{E}} \leq \gamma \|x - y\|_{c,\bar{E}}$.

2. Let $w^t := \hat{H}(x^t) - H(x^t)$ denote the stochastic error associated with $\hat{H}$ at iteration $t$ and let $\mathcal{F}^t := \{x^1, \ldots, x^t\}$ denote the filtration up to time $t$. Then $w^t$ satisfies the following properties,

   - *Martingale noise:* $\mathbb{E}[w^t | \mathcal{F}^t] = 0$.
   - *Bounded variance:* $\mathbb{E}[\|w^t\|_{c,\bar{E}}^2 | \mathcal{F}^t] \leq A + B \|x^t - x^*\|_{c,\bar{E}}^2$ for some fixed constants $A$ and $B$.

3. *There exist a fixed equivalent class, i.e., $x^*$ for which $\|H(x^*) - x^*\|_{c,\bar{E}} = 0$.*

We begin by analyzing the behavior of $M_{\bar{E}}$ for a fixed $t$ using its L-smoothness property shown in Proposition 2:

$$M_{\bar{E}}(x^{t+1} - x^*) \leq M_{\bar{E}}(x^t - x^*) + \langle \nabla M_{\bar{E}}(x^t - x^*), x^{t+1} - x^t \rangle + \frac{L}{2} \|x^{t+1} - x^t\|_{s,\bar{E}}^2. \tag{B.6}$$

First, we show the linear term above induces a negative drift.

**Lemma 5.** *Let $M_{\bar{E}}$ be defined in (B.3). Then conditioned on $\mathcal{F}^t$, $x^{t+1}$ satisfies*

$$\mathbb{E}[\langle \nabla M_{\bar{E}}(x^t - x^*), x^{t+1} - x^t \rangle] \leq -2\beta\eta_t M_{\bar{E}}(x^t - x^*),$$

*with $\beta \geq (1 - \gamma\sqrt{c_u/c_l})$, where $c_u, c_l$ are the uniform approximation parameters of $M$ defined in (B.1).*

*Proof.* First, due to the martingale noise assumption for $\hat{H}$, the following relation holds conditioned on $\mathcal{F}^t$,

$$\mathbb{E}[\langle \nabla M_{\bar{E}}(x^t - x^*), x^{t+1} - x^t \rangle] = \eta_t \mathbb{E}[\langle \nabla M_{\bar{E}}(x^t - x^*), H(x^t) - x^t + w^t \rangle] = \eta_t \langle \nabla M_{\bar{E}}(x^t - x^*), H(x^t) - x^t \rangle.$$

Now we study the last term. The convexity of $M_{\bar{E}}$ implies that

$$\begin{aligned}
\langle \nabla M_{\bar{E}}(x^t - x^*), H(x^t) - x^t \rangle &= \langle \nabla M_{\bar{E}}(x^t - x^*), H(x^t) - x^* + x^* - x^t \rangle \\
&\leq M_{\bar{E}}(H(x^t) - x^*) - M_{\bar{E}}(x^t - x^*) \\
&\overset{(a)}{\leq} \frac{1}{2c_l} \|H(x^t) - H(x^*)\|_{c,\bar{E}}^2 - M_{\bar{E}}(x^t - x^*) \\
&\overset{(b)}{\leq} \frac{\gamma^2}{2c_l} \|x^t - x^*\|_{c,\bar{E}}^2 - M_{\bar{E}}(x^t - x^*) \\
&\leq (\frac{\gamma^2 c_u}{c_l} - 1) M_{\bar{E}}(x^t - x^*) \leq -(1 - \gamma\sqrt{c_u/c_l}) M_{\bar{E}}(x^t - x^*),
\end{aligned}$$

where $(a)$ follows from $x^*$ belonging to a fixed equivalent class with respect to $H$ and $(b)$ follows from the contraction property of $H$. $\qquad\square$

Now let's focus on the last term in (B.6). In [18], the authors utilize norm equivalence to upper bound $\|x\|_s^2$ by some $l_s \|x\|_c^2$ so that it could be bounded by $M$. We apply the same technique in the next lemma. Notice that the monotonicity of infimal convolution (Lemma 4.a) and Lemma 4.b)) implies that $\|x\|_{s,\bar{E}}^2 \leq l_s \|x\|_{c,\bar{E}}^2$.

**Lemma 6.** *If* $\|x\|_{s,\bar{E}}^2 \le l_s \|x\|_{c,\bar{E}}^2$, *then conditioned on* $\mathcal{F}^t$, $x^{t+1}$ *generated by* (B.5) *satisfies*

$$\mathbb{E}[\|x^{t+1} - x^t\|_{s,\bar{E}}^2] \le (16 + 4B)c_u l_s \eta_t^2 M_{\bar{E}}(x^t - x^*) + 2A l_s \eta_t^2.$$

*Proof.* By update rule (B.5), we have

$$\begin{aligned}
\mathbb{E}[\|x^{t+1} - x^t\|_{s,\bar{E}}^2] &= \eta_t^2 \mathbb{E}[\|H(x^t) + w^t - x^t\|_{s,\bar{E}}^2] \\
&\overset{(a)}{\le} 2\eta_t^2 \mathbb{E}[\|H(x^t) - x^t\|_{s,\bar{E}}^2 + \|w^t\|_{s,\bar{E}}^2] \\
&\le 2\eta_t^2 l_s \mathbb{E}[\|H(x^t) - x^t\|_{c,\bar{E}}^2 + \|w^t\|_{c,\bar{E}}^2] \\
&\le 2\eta_t^2 l_s \mathbb{E}[2\|H(x^t) - H(x^*)\|_{c,\bar{E}}^2 + 2\|x^t - x^*\|_{c,\bar{E}}^2 + \|w^t\|_{c,\bar{E}}^2] \\
&\le \eta_t^2 l_s (8 + 2B)\|x^t - x^*\|_{c,\bar{E}}^2 + \eta_t^2 l_s 2A \\
&\overset{(b)}{\le} \eta_t^2 l_s c_u (16 + 4B) M_{\bar{E}}(x^t - x^*) + \eta_t^2 l_s 2A,
\end{aligned}$$

where (a) follows from the triangle inequality and (b) follows from the uniform approximation property of $M_{\bar{E}}$. $\qquad\square$

Putting them together, we get the following recursive relation.

**Proposition 3.** *Let* $x^t$ *be generated by* (B.5) *using* $\hat{H}$ *satisfying Assumption 4 and let* $\|x\|_{s,\bar{E}}^2 \le l_s \|x\|_{c,\bar{E}}^2$, $\forall x$. *Then the following relation holds conditioned on* $\mathcal{F}^t$,

$$\mathbb{E}[M_{\bar{E}}(x^{t+1} - x^*)] \le (1 - 2\alpha_2 \eta_t + \alpha_3 \eta_t^2) M_{\bar{E}}(x^t - x^*) + \alpha_4 \eta_t^2, \tag{B.7}$$

*where* $\alpha_2 := (1 - \gamma\sqrt{c_u/c_l})$, $\alpha_3 := (8 + 2B)c_u l_s L$, $\alpha_4 := A l_s L$.

*Proof.* By substituting Lemma 5 and 6 into (B.6), we get $\mathbb{E}[M_{\bar{E}}(x^{t+1} - x^*)] \le (1 - 2\beta\eta_t + (8 + 2B)c_u l_s L \eta_t^2) M_{\bar{E}}(x^t - x^*) + A l_s \eta_t^2$. $\qquad\square$

Next, we suggest a specific stepsize $\eta_t$ to calculate the convergence rate.

**Theorem 3.** *Let* $\alpha_2$, $\alpha_3$ *and* $\alpha_4$ *be defined in 3. If* $x^t$ *is generated by* (B.5) *with an* $\hat{H}$ *satisfying Assumption 4 and stepsizes* $\eta_t := \frac{1}{\alpha_2(t+K)}$, $K := \max\{\alpha_3/\alpha_2, 3\}$,

$$\mathbb{E}[\|x^N - x^*\|_{c,\bar{E}}^2] \le \frac{K^2}{(N+K)^2}\frac{c_u}{c_l}\|x^0 - x^*\|_{c,\bar{E}}^2 + \frac{8\alpha_4 c_u}{(N+K)\alpha_2^2}, \ \forall N \ge 1. \tag{B.8}$$

*Else if a constant stepsize eta with* $\eta_t \alpha_3/\alpha_2 \le 1$, *then*

$$\mathbb{E}[\|x^N - x^*\|_{c,\bar{E}}^2] \le \frac{c_u}{c_l}(1 - \alpha_2)^N \|x^0 - x^*\|_{c,\bar{E}} + \frac{c_u \alpha_4}{\alpha_2}\eta, \ \forall N \ge 1. \tag{B.9}$$

*Proof.* Let's consider the decreasing stepsize first. Since $\eta_t$ satisfies $\alpha_3 \eta_t^2 \le \alpha_2 \eta_t$, it follows from (B.7) that

$$\mathbb{E}[M_{\bar{E}}(x^{t+1} - x^*)] \le (1 - \alpha_2 \eta_t) M_{\bar{E}}(x^t - x^*) + \alpha_4 \eta_t^2.$$

By letting $\Gamma_t := \prod_{i=0}^{t-1}(1 - \alpha_2 \eta_t)$, we can obtain the $N$-step recursion relationship

$$\mathbb{E}[M_{\bar{E}}(x^{t+1} - x^*)] \le \Gamma_N M_{\bar{E}}(x^t - x^*) + \frac{\alpha_4}{\alpha_2}\Gamma_N \sum_{t=0}^{N-1}(\frac{1}{\Gamma_{t+1}})\alpha_2 \eta_t^2.$$

Then the algebraic relationship $\frac{1}{\Gamma_{t+1}}(\alpha_2 \eta_t) = \frac{1}{\Gamma_{t+1}} - \frac{1}{\Gamma_t}$ implies that

$$\mathbb{E}[M_{\bar{E}}(x^{t+1} - x^*)] \le \Gamma_N M_{\bar{E}}(x^t - x^*) + \frac{\alpha_4}{\alpha_2}\Gamma_N \sum_{t=0}^{N-1}(\frac{1}{\Gamma_{t+1}} - \frac{1}{\Gamma_t})\eta_t.$$

Moreover a careful computation shows that

$$\Gamma_t = \frac{(K-1)(K-2)}{(t+K-1)(t+K-2)}, \Gamma_N \le \frac{K^2}{(N+K)^2}, \Gamma_N \sum_{t=0}^{N-1}\eta_t(\frac{1}{\Gamma_{t+1}} - \frac{1}{\Gamma_t}) \le \frac{4}{\alpha_2(N+K)}.$$

Thus we can conclude (B.8) by noting that $M_{\bar{E}}$ is an uniform approximation of $\|\cdot\|_{c,\bar{E}}$, i.e., $c_l M_{\bar{E}}(x) \leq \frac{1}{2}\|x\|_{c,\bar{E}}^2 \leq c_u M_{\bar{E}}(x)$.

Next, for the constant stepsize, again, we can recover from (B.7) that

$$\mathbb{E}[M_{\bar{E}}(x^{t+1} - x^*)] \leq (1 - \alpha_2 \eta) M_{\bar{E}}(x^t - x^*) + \alpha_4 \eta^2, i.e.,$$

$$\mathbb{E}[M_{\bar{E}}(x^N - x^*)] \leq (1 - \alpha_2 \eta)^N M_{\bar{E}}(x^t - x^*) + \alpha_4 \eta^2 \sum_{t=0}^{N-1}(1 - \alpha_2 \eta)^t \leq (1 - \alpha_2 \eta)^N M_{\bar{E}}(x^t - x^*) + \frac{\alpha_4}{\alpha_2}\eta,$$

from which (B.9) follows naturally. $\qquad\square$

## B.3 Convergence of the $J$-step $Q$-learning Algorithm

We establish the convergence of the $J$-step $Q$-learning algorithm in this subsection. With $\bar{E} := \{ce : c \in \mathbb{R}\}$, the sample $J$-step Bellman operator $\hat{H}^J$ satisfies Assumption 4:

1. $H^J$ is $\gamma$-contractive with respective to span infinite norm with $0 < \gamma < 1$ i.e., $\left\| H^J(Q) - H^J(\bar{Q}) \right\|_{\infty,\bar{E}} \leq \gamma \left\| Q - \bar{Q} \right\|_{\infty,E}$.

2. Let $w^t := \hat{H}^J(Q_t) - H^J(Q_t)$ denote the stochastic error associated with $\hat{H}^J$ at iteration $t$ and let $\mathcal{F}^t := \{Q_1, \ldots, Q_t\}$ denote the filtration up to time t. Then $w^t$ satisfies the following properties,

   - Martingale noise: $\mathbb{E}[w^t | \mathcal{F}^t] = 0$.
   - Bounded variance: $\mathbb{E}[\|w^t\|_{\infty,E}^2 | \mathcal{F}^t] \leq \underbrace{2(J^2 + \|Q^*\|_{\infty,\bar{E}}^2)}_{A} + \underbrace{2}_{B}\|Q_t - Q^*\|_{\infty,\bar{E}}^2$.

3. There exists a gain optimal $Q^*$ for which $\left\| \hat{H}^J(Q^*) - Q^* \right\|_{\infty,E} = 0$.

We choose the following $l_\infty$-norm smoothing function introduced in [18] as our base Lyapunov function

$$M(x) := \tfrac{1}{2}(\|\cdot\|_\infty^2 \,\square\, \frac{1}{\mu}\|\cdot\|_{4\log|S||A|}^2), \text{ with } \mu = (\tfrac{1}{2} + \tfrac{1}{2\gamma})^2 - 1.$$

Then the following problem parameters for analyzing the convergence of the SA scheme can be derived:

$$c_u = (1 + \mu), c_l = (1 + \mu/\sqrt{e}), L = \tfrac{4\log|S||A|}{\mu}, l_s = \sqrt{e}.$$

Following the same algebraic manipulation in Section A.6 of [18], we get

$$\alpha_1 = c_u/c_l \leq \sqrt{e} \leq \tfrac{3}{2},$$
$$\alpha_2 = (1 - \gamma\sqrt{c_u/c_l}) \geq 1 - \gamma(1 + \mu)^{1/2} = \tfrac{1-\gamma}{2},$$
$$\alpha_3 = (8 + 2B)c_u l_s L = 12\tfrac{1+\mu}{\mu}4\log(|S||A|)\sqrt{e} \leq \tfrac{144}{(1-\gamma)}\log(|S||A|),$$
$$\alpha_4 c_u = A c_u l_s L \leq 2(J^2 + \|Q^*\|_{\infty,\bar{E}}^2)\tfrac{1+\mu}{\mu}4\log(|S||A|) \leq \tfrac{24\log(|S||A|)}{(1-\gamma)}(J^2 + \|Q^*\|_{\infty,\bar{E}}^2).$$

Then the exact convergence rate of Algorithm 2 can obtained by merely substituting them into Theorem 3. And the convergence and sample complexity Theorem 2 in the main text is a simple corollary of the next result.

*Proof.* **Proof of Theorem 2:** The result follows from merely substituting the above estimates into (B.9) and (B.8). In particular, the following conservative estimates are used for calculation

$$\alpha_1 = \tfrac{3}{2}, \alpha_2 = \tfrac{1-\gamma}{2}, \alpha_3 = \tfrac{144}{(1-\gamma)}\log(|S||A|) \text{ and } \alpha_4 c_u \leq \tfrac{24\log(|S||A|)}{(1-\gamma)}(J^2 + \|Q^*\|_{\infty,\bar{E}}^2).$$

$\qquad\square$

# C   Implementation Detail for Numerical Experiments

## C.1   Setup

We consider an MRP with $|\mathcal{S}| = 100$ states, where rewards and transition probabilities are generated as follows:

Rewards: The reward $\mathcal{R}(s)$ for each state is drawn from the uniform distribution on $[0, 1]$.

Transition probabilities: For each state $s \in \mathcal{S}$, the transition probabilities $P(s, s')$ to each successor state $s' \in \mathcal{S}$ are chosen as random partitions of the unit interval. That is, $|\mathcal{S}| - 1$ numbers are chosen uniformly randomly between 0 and 1, dividing that interval into $|\mathcal{S}|$ numbers that sum to one – the probabilities of the $|\mathcal{S}|$ successor states.

We first compute the stationary distribution $\pi$ of the MRP, and then obtain the average-reward $r^* := \pi^\top \mathcal{R}$, and the basic differential value function $v^*$ by solving the following linear system of equations:

$$(I - P)\, v^* = \mathcal{R} - r^* e \text{ and } \pi^\top v^* = 0.$$

For linear function approximation, we consider a feature matrix $\Phi$ with $d = 20$ features for each state $s \in \mathcal{S}$. We first generate a matrix $\tilde{\Phi} \in \mathbb{R}^{|\mathcal{S}| \times (d-2)}$, where each element is drawn from the Bernoulli distribution with success probability $p = 0.5$. Then, we construct $\Phi \in \mathbb{R}^{|\mathcal{S}| \times d}$ by stacking the all-ones vector $e$ and the basic differential value function $v^*$ as columns into the the matrix $\tilde{\Phi}$, i.e., $\Phi := \begin{bmatrix} \tilde{\Phi} & e & v^* \end{bmatrix}$. We repeat this process until we obtain a full column rank feature matrix. We further normalize the features to ensure $\|\phi(s)\| \leq 1$ for all $s \in \mathcal{S}$. With the above feature matrix, we can easily compute $\theta_e$ and $\theta^*$ by solving

$$\Phi \theta_e = e \text{ and } \Phi \theta^* = v^*.$$

## C.2   1st Experiment

In the first experiment, we show that the iterates $\theta_t$ of Alorithm 1 converge to different TD limit points when the initial points $\theta_0$ are different. We set $\lambda = 0$, $c_\alpha = 1$, $T = 100,000$, $\beta_t = \frac{150}{t+1000}$ and $\bar{r}_0 = 0$. We draw 4 $d$-dimensional vectors from the uniform distribution with lower bound $= -5$ and upper bound $= 5$. We then use each of the samples as the initial guess $\theta_0$, and plot $\mathbb{E}\left[\|\Pi_{2,E}(\theta_t - \theta^*)\|_2\right]$ and $\mathbb{E}\left[(\theta_t - \theta^*)^\top \frac{\theta_e}{\|\theta_e\|_2}\right]$ in Figure 1 and 2. Note that, each curve is average over 100 independent runs with the same $\theta_0$.

## C.3   2nd Experiment

In the second experiment, we empirically verify the performance upper bounds of Alorithm 1 in Theorem 1. We set $c_\alpha = 1$, $T = 1,000,000$, $\beta_t = \frac{150}{t+1000}$, $\bar{r}_0 = 0$ and $\theta_0 = 0$ and consider $\lambda \in \{0, 0.2, 0.4, 0.8\}$. In Figure 3, we plot $\mathbb{E}\left[(\bar{r}_t - r^*)^2 + \|\Pi_{2,E}(\theta_t - \theta^*)\|_2^2\right]$ as a function of $t$ for $t \in [0, 10^5)$, and in Figure 4, we plot $\ln \mathbb{E}\left[(\bar{r}_t - r^*)^2 + \|\Pi_{2,E}(\theta_t - \theta^*)\|_2^2\right]$ as a function of $\ln t$ for $t \in [5 \times 10^5, 10^6)$. Each curve is average over 100 independent runs with the same $\lambda$.

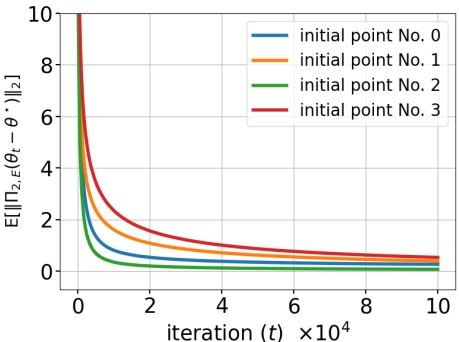

Figure 1: Convergence of the iterates $\theta_t$ to the set of TD limit points for $4$ different initial points.

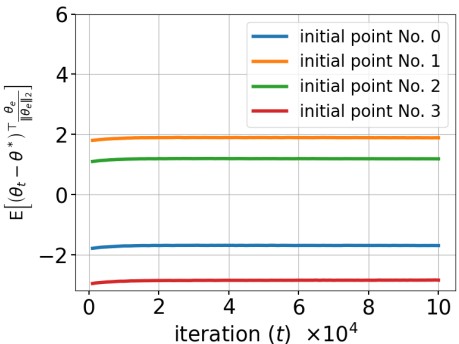

Figure 2: Convergence of the projection of the iterates $\theta_t$ onto the set of TD limit points for $4$ different initial points.

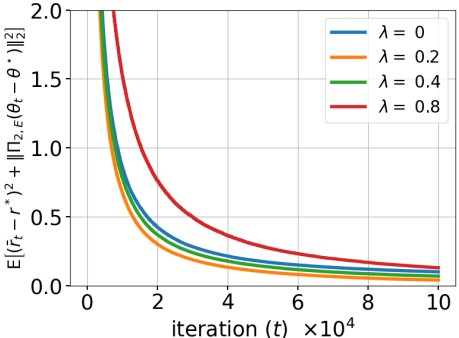

Figure 3: Convergence of the iterates $(\bar{r}_t, \theta_t)$ for $\lambda \in \{0, 0.2, 0.4, 0.8\}$.

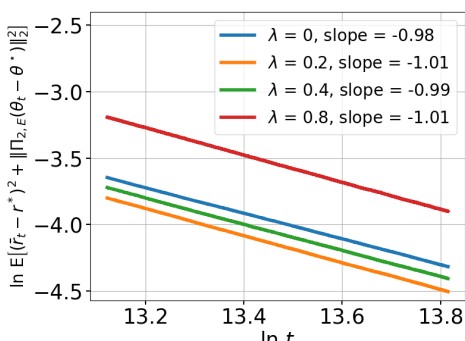

Figure 4: Asymptotic convergence rate of the iterates $(\bar{r}_t, \theta_t)$ for $\lambda \in \{0, 0.2, 0.4, 0.8\}$.