# OpenReview forum: "Finite Sample Analysis of Average-Reward TD Learning and $Q$-Learning"
_NeurIPS.cc/2021/Conference — NeurIPS 2021 Poster_

### Official Review · Reviewer_qoyS · 2021-07-11

**Rating:** 7
**Confidence:** 4

**Summary:**

This paper focuses on the infinite-horizon average-reward setting and develops the non-asymptotic upper bound for TD($\lambda$) with linear function approximation and tabular Q-learning. Its upper bounds match the best-known results on the discounted-reward MDP for $\epsilon$-accuracy. Its technique used to deal with the non-uniqueness of the limit point seems new, and its construction of Lyapunov function from noticing the connection between the span seminorm and the infimal convolution is a non-trivial observation.

**Limitations And Societal Impact:**

Since this paper is purely theoretical, the limitations and societal impact section are not applicable.

**Main Review:**

Thanks for this well-written work with solid theoretical analysis on the infinite-horizon average-reward RL problem. I agree that this paper has its own technical novelty in dealing with the non-uniqueness of limit points. All derived upper bounds nearly match the existing infinite-horizon discounted-reward RL problem. There are several of my concerns regarding this paper:
1.	I am not clear if the average-reward setting is practical in real-life applications since most of the cited applications ([2], [3], and [4]) are from 20 years ago. Is there any more recent work using this setup?
2.	Assumption 3 in Section 4.3 seems unnatural for me. For the traditional discounted-reward setup, there is no such assumption. Does this assumption automatically hold for discounted-reward MDP?
3.	From Theorem 2, it seems that $\gamma$ is comparable with the discounted factor. However, the dependence on $\gamma$ is worse than the best-known result on discounted-reward Q-learning: In the paper below, the order of $1/(1-\gamma)$ is 4 rather than 5.
Li, Gen, et al. "Is Q-learning minimax optimal? a tight sample complexity analysis." arXiv preprint arXiv:2102.06548 (2021).
Moreover, the analysis of tabular Q-learning is not as complete as TD($\lambda$). In Section 3.3, this paper introduces two learning rate schedules: constant learning rate and decaying learning rate. But for tabular Q-learning, this paper only considers the decaying learning rate. So, I think there are still many works that should be done on the Q-learning section.

In summary, this paper can be further improved by introducing more recent applications and adding more explanations on the non-standard assumption. Regarding the Q-learning section, I don’t think it is complete enough, as stated above, so I cannot give a clear accept.

**Time Spent Reviewing:**

10

---

> ### Author Response · Authors · 2021-08-11
> **Responses to Reviewer qoyS**
>
> We thank the reviewer for the valuable feedback. Please find our responses below:
>
> **(Main Review)[Concern 1.]** "I am not clear if the average-reward setting is practical in real-life applications since most of the cited applications ([2], [3], and [4]) are from 20 years ago. Is there any more recent work using this setup?"
>
> **[Response]** There are more recent works considering the average-reward setting, for example,
>
> [[Zhang & Ross, 2021](https://arxiv.org/pdf/2106.07329.pdf)] develops the average-reward trust region policy optimization (TRPO) algorithm and demonstrates that it significantly outperforms TRPO (its discounted counterpart) in the most challenging MuJuCo [[Todorov et al., 2012](https://ieeexplore.ieee.org/stamp/stamp.jsp?arnumber=6386109&casa_token=5lMgsNSA4S8AAAAA:AuDF_FnDRBMHcyPFfiCE06RkB-mrMozzKSKKMM_54YMhMfW-YoMNJQA0-LS_IKsF2jrx1Ep1sQ&tag=1)] environments.
>
> [[Dai & Gluzman, 2020](https://arxiv.org/pdf/2008.01644.pdf)] considers the multiclass queueing networks control problems with long-run average cost objectives. They develop the average-reward proximal policy optimization algorithm and demonstrate that it can generate control policies that consistently beat the performance of all state-of-arts control policies known in the literature.
>
> We note that a non-discounted performance measure is used to evaluate the trained policies in many benchmarks for reinforcement learning algorithms such as Atari games [[Mnih et al., 2013](https://arxiv.org/pdf/1312.5602.pdf)] and MuJoCo. Most modern Deep RL algorithms introduce a discount factor during training even when the natural objective of interest is undiscounted. This leads to a discrepancy between the evaluation and training objective.
>
> **(Main Review)[Concern 2.]** "Assumption 3 in Section 4.3 seems unnatural for me. For the traditional discounted-reward setup, there is no such assumption. Does this assumption automatically hold for discounted-reward MDP?"
>
> **[Response]** The Bellman operator in discounted-reward MDPs is most well-known to be a contraction mapping with respect to supremum norm, with contraction factor being the discount factor. This is the property that is crucially exploited in most of the analysis of discounted-reward MDP or RL algorithms. However, a less well-known property of the Bellman operator in the discounted setting is that it is also a contraction with respect to the span seminorm with contraction factor being upper bounded by the discounted factor (please see Theorem 6.6.6 in [[Puterman, 2014](https://scholar.google.com/citations?view_op=view_citation&hl=en&user=uCCMYzIAAAAJ&citation_for_view=uCCMYzIAAAAJ:u5HHmVD_uO8C)] for the proof). Thus, for discounted-reward MDPs, Assumption 3 in section 4.3 holds automatically with the contraction factor $\gamma$ being the discount factor and $J=1$.
>
> Even classical analysis of average-reward MDPs such as in [[Puterman, 2014](https://scholar.google.com/citations?view_op=view_citation&hl=en&user=uCCMYzIAAAAJ&citation_for_view=uCCMYzIAAAAJ:u5HHmVD_uO8C); [Bertsekas, 2012](https://scholar.google.com/citations?view_op=view_citation&hl=en&user=VUmcVOAAAAAJ&citation_for_view=VUmcVOAAAAAJ:d1gkVwhDpl0C)]
> is more challenging than that of the discounted-reward MDPs as the Bellman operator is no longer a contraction mapping with respect to any norm. Therefore, Assumption 3 arises only when  dealing with average-reward problem. However, Assumption 3 is justified because it is shown in Section 8.5.4 of [[Puterman, 2014](https://scholar.google.com/citations?view_op=view_citation&hl=en&user=uCCMYzIAAAAJ&citation_for_view=uCCMYzIAAAAJ:u5HHmVD_uO8C)] that any unichain MDP can be made to satisfy this assumption by an aperiodicity transformation.
>
> **(Main Review)[Concern 3.]** "From Theorem 2, it seems that $\gamma$ is comparable with the discounted factor. However, the dependence on  is worse than the best-known result on discounted-reward $Q$-learning: In the paper below, the order of $1/(1-\gamma)$ is 4 rather than 5. Li, Gen, et al. "Is Q-learning minimax optimal? a tight sample complexity analysis." arXiv preprint arXiv:2102.06548 (2021). Moreover, the analysis of tabular Q-learning is not as complete as TD($\lambda$). In Section 3.3, this paper introduces two learning rate schedules: constant learning rate and decaying learning rate. But for tabular Q-learning, this paper only considers the decaying learning rate. So, I think there are still many works that should be done on the Q-learning section."
>
> **[Response]** For the purpose of this comment, we use  $\tilde{\gamma}$ to denote the discount factor of a discounted MDP.  The contraction factor of the Bellman operator in the discounted setting is the discount factor $\tilde{\gamma}$, and the finite time bounds inevitably depend on the factor $1/(1-\tilde{\gamma})$. This factor plays a significant role for the following reason.
> In a discounted MDP, since future rewards are discounted, they become irrelevant after some time. This time horizon is approximately $1/(1-\tilde{\gamma})$. For any RL problem, since one wants to learn over a long time-horizon, it is imperative that the discount factor $\tilde{\gamma}$ is picked close to 1, so that the effective horizon  $1/(1-\tilde{\gamma})$  is large leading to a large factor in the finite-time bounds. An alternate way of thinking of the same issue is the following.
> While many learning problems in practice are interested in infinite-horizon average rewards, one uses the discounted-reward model to learn them because of its simplicity. However, a well-known result from MDP theory [[Puterman, 2014](https://scholar.google.com/citations?view_op=view_citation&hl=en&user=uCCMYzIAAAAJ&citation_for_view=uCCMYzIAAAAJ:u5HHmVD_uO8C)] states that the optimal policy of the average-reward problem is the same as the optimal policy of the discounted-reward problem, when the discount factor  $\tilde{\gamma}$ approaches 1. Therefore, the factor $1/(1-\tilde{\gamma})$  plays an important role in discounted-reward analysis, since the regime of interest is when $\tilde{\gamma}$ is close to  1.
>
> In contrast, the term $1/(1-\gamma)$ in the average-reward setting does not have such an interpretation as the effective horizon. It simply shows up due to  $\gamma$  being the span contraction factor in Assumption 3. The effective horizon in the average-reward setting is simply infinite. So, the dependence on $1/(1-\gamma)$  in the finite-time bounds in the average-reward setting is not as significant as in the discounted setting. We agree with the reviewer that by utilizing the modified analysis in [[Li et al., 2021](https://arxiv.org/pdf/2102.06548.pdf)] the dependence might be improved to $1/(1-\gamma)^4$, which is a future research direction.
>
> We omitted the constant stepsize only for brevity. Under the proper choice of the constant stepsize $\epsilon > 0$, a finite- sample bound for $\mathbf{E}[\frac{1}{2}||Q_t - Q^*||^2_{\infty, sp}]$ of the form $O(\{\alpha^t \cdot (\text{deterministic error}) + \epsilon \cdot (\text{stochastic error})\})$, where $\alpha<1$ depends on $\epsilon$, can be derived as an immediate consequence of Proposition 3 in Appendix B.  The detailed convergence bound will be included in the camera-ready version.

---

> > ### Comment · Reviewer_qoyS · 2021-08-17
> > **All concerns are addressed**
> >
> > Thanks! All my concerns are addressed. I am convinced that the dependence on $1/(1-\gamma)$ is not very significant in the average-reward setting; therefore, I think the theoretical results provided in this paper are asymptotically tight enough.
> >
> > Since this paper is the first work considering the non-uniqueness of limiting points and contains some specific analysis on this challenge, I will raise my score from 6 to 7.

---

### Official Review · Reviewer_ZKyN · 2021-07-15

**Rating:** 6
**Confidence:** 4

**Summary:**

This paper studies the average reward TD-learning and Q-learning. In detail, the authors analyze the TD-$\lambda$ learning with linear function approximation. A finite sample analysis is given for the constant step size and diminishing step size. They also provide a finite-sample analysis of Q-learning in a tabular setting. The main contribution in this paper is how to deal with the constant shift in Bellman equality. In detail, in the TD-learning part, their analysis does not need to assume the uniqueness of the limit point of TD-$\lambda$. In the Q-learning part, they build the contraction based on the distance between two function classes to consider the constant shift.

**Limitations And Societal Impact:**

There's no negative societal impact in this paper.

Suggestions:

The paper is overall easy to follow.  In Corollary 1, I'll suggest the authors restate that the learning rate is set as the case (b) in Theorem 1. It confused me at first because it is not well-clarified.

Furthermore, there's a series of works [2, 3] also providing finite-time analysis in the average reward setting using Actor-Critic with linear function approximation. However, all these works require the unique limit point of TD. Therefore, it would be interesting if the authors can comment about how the proposed method in this paper can be extended to these works to remove such uniqueness assumption. These comments will help readers understanding the contribution of this paper better.

[2] Qiu, Shuang, et al. "On Finite-Time Convergence of Actor-Critic Algorithm." IEEE Journal on Selected Areas in Information Theory 2.2 (2021): 652-664.

[3] Wu, Yue, et al. "A finite time analysis of two time-scale actor critic methods." Advances in Neural Information Processing Systems, 2020.


**Main Review:**

This paper is well organized and written. It provides a solid theoretical tool to analyze the constant shift in the value function. However, for the TD-learning part, the proof technique is similar to [1]. The result, including the constant step size and the diminishing step size, is also similar to [1]. Therefore, the contribution of this paper is somehow limited, and the result cannot be considered groundbreaking.

Another drawback in the Q-learning part is the proposed algorithm needs a generative model which allows simulating the trajectory for $J$ step given arbitrary initial state and action $(s, a)$. This will make the algorithm impractical, comparing to an algorithm receiving the initial state from an unknown distribution.

I tend to suggest marginally reject this paper due to the limitation of contribution. I'd like to change my adjust my rating after discussion.

[1] Bhandari, Jalaj, Daniel Russo, and Raghav Singal. "A finite time analysis of temporal difference learning with linear function approximation." Conference on learning theory. PMLR, 2018.

**Time Spent Reviewing:**

6

---

> ### Author Response · Authors · 2021-08-11
> **Responses to Reviewer ZKyN**
>
> We thank the reviewer for the valuable feedback. Please find our responses below:
>
> **(Main Review)[Comment 1.]**  "This paper is well organized and written. It provides a solid theoretical tool to analyze the constant shift in the value function. However, for the TD-learning part, the proof technique is similar to [1]. The result, including the constant step size and the diminishing step size, is also similar to [1]. Therefore, the contribution of this paper is somehow limited, and the result cannot be considered groundbreaking."
>
> **[Response]** Our analysis of $Q$-learning is not related to that of [Bhandari et al., 2018], and crucially relies on the construction of a Lyapunov function. Our analysis of TD is in same spirit as that of [Bhandari et al., 2018], but with two differences that we highlight in the following:
>
> 1. The crucial steps of the analysis of [[Bhandari et al., 2018](https://arxiv.org/pdf/1806.02450.pdf)] leverage the properties of the TD limit point established in [[Tsitsiklis & Van Roy, 1997](https://ieeexplore.ieee.org/stamp/stamp.jsp?arnumber=580874&casa_token=ZpC5s1Pty58AAAAA:aPdXydVmwPqpdEX6O_ZBPHDdfxLPbaJkb9v-gil_zKxCvbTTdc9XIdMyGb0T2C9YRBSovTGMvw&tag=1)] and the contraction property of the projected Bellman operator. In contrast, the projected Bellman operator is *not* a contraction in the average-reward setting and the properties of the TD fixed points are not well established in the literature. We handle the non-uniqueness of the TD fixed point in our analysis, which is assumed away in previous works, and establish the properties of the TD fixed point and the projected Bellman equation.
>
> 2. Moreover, we have a crucial technical difference from [[Bhandari et al., 2018](https://arxiv.org/pdf/1806.02450.pdf)]. An additional projection step (onto a large enough ball) is needed for the analysis of [[Bhandari et al., 2018](https://arxiv.org/pdf/1806.02450.pdf)] to maintain the boundedness of the iterates. Such a projection step is not practical in RL because one needs to know the problem parameters to pick the projection set so that the TD fixed point lies in it. We provide finite-sample convergence guarantees without needing such an impractical projection step.
>
> The fact that convergence results under constant and diminishing step-sizes look similar to [[Bhandari et al., 2018](https://arxiv.org/pdf/1806.02450.pdf)] do not diminish our contribution, because it is well-known that convergence results of stochastic approximation or SGD all have similar qualitative behavior [[Kushner & Yin 2003](https://scholar.google.com/citations?view_op=view_citation&hl=en&user=nlZ0o_4AAAAJ&citation_for_view=nlZ0o_4AAAAJ:u5HHmVD_uO8C); [Borkar, 2009](https://scholar.google.com/citations?view_op=view_citation&hl=en&user=Km1V8WwAAAAJ&citation_for_view=Km1V8WwAAAAJ:u5HHmVD_uO8C)]. The key challenge is to obtain these results for the specific setting under consideration. In our case, the key difficulty is overcoming the lack of contraction property, which is crucially exploited in the analysis of discounted-reward problems.
>
> **(Main Review)[Comment 2.]** "Another drawback in the $Q$-learning part is the proposed algorithm needs a generative model which allows simulating the trajectory for $J$ step given arbitrary initial state and action $(s,a)$. This will make the algorithm impractical, comparing to an algorithm receiving the initial state from an unknown distribution."
>
> **[Response]** Indeed, a limitation of our current result on $Q$-learning is that we consider synchronous $Q$-learning which cannot be implemented using a single trajectory. However, there was no known prior result on average-reward $Q$-learning even in the synchronous setting due to the lack of the contraction property. We are the first one to obtain the finite-sample bounds, which we do by exploiting the span seminorm contraction property. Since it is well-known that the average-reward problems are more challenging than their discounted-reward counterparts, we believe that this itself is a significant contribution. We believe that building upon our analysis, future work will consider the asynchronous setting. Results in the discounted setting were obtained in a similar step-by-step procedure. Finite-sample bounds for the synchronous version were first obtained in [[Chen et al., 2020](https://arxiv.org/pdf/2002.00874.pdf)] and [[Wainwright, 2019](https://arxiv.org/pdf/1905.06265.pdf)], and these breakthrough results later led to similar bounds for asynchronous version [[Qu & Wierman, 2020](http://proceedings.mlr.press/v125/qu20a/qu20a.pdf); [Chen et al., 2021](https://arxiv.org/pdf/2102.01567.pdf)].
>
> We now present an approach one can take to study the asynchronous version, and highlight the key technical challenge. Because of the $J$-step Bellman operator $H^J$ (introduced in section 4.2, line 256), one can use the following asynchronous sampling scheme. Given a behavior policy $\pi$, an arbitrary starting point $s^0$ and an initial value function $Q_0$, the following trajectory can be drawn:
> $$(s^0,\pi(s^0)) \rightarrow (s^1, \mu_{Q_0}(s^1)) \rightarrow \ldots \rightarrow (s^{J-1}, \mu_{Q_0}(s^{J-1})) \rightarrow (s^{J}, \pi(s^{J})) \rightarrow (s^{J+1}, \mu_{Q_1}(s^{J+1})) \rightarrow \ldots \rightarrow (s^{2J-1}, \mu_{Q_1}(s^{2J-1})) \rightarrow (s^{2J}, \pi(s^{2J})) \rightarrow (s^{2J+1}, \mu_{Q_2}(s^{2J+1})) \rightarrow (s^{2J+2}, \mu_{Q_2}(s^{2J+2})) \rightarrow \ldots \rightarrow (s^{3J - 1}, \mu_{Q_2}(s^{3J-1})) \rightarrow (s^{3J}, \pi(s^{3J})) \ldots,$$
>   where $\mu_{Q}$ denotes the greedy policy with respect to the value function $Q$.
>   Since $\mu_{Q_0}$ is fixed during the first $J$ steps, the sample estimate $\mathcal{R}(s^0,\pi(s^0)) + \sum_{j=1}^{J-1} \mathcal{R}(s^j, \mu_{Q_0}(s^j))$, being an unbiased estimator of $H^{J}(Q_0)(s^0,\pi(s^0)),$ can be used to update $Q_1(s^0,\pi(s^0))$ in the same way as Algorithm 2. Analyzing such an algorithm is especially challenging because the underlying stochastic process is a time varying Markov Chain; the $\mu_Q$ policy is changed every $J$-step.
> This challenge can be overcome by using recent techniques that are used to study algorithms with time-varying Markov chains [[Khodadadian et al., 2021](https://arxiv.org/pdf/2101.10506.pdf); [Wu et al., 2020](https://arxiv.org/pdf/2005.01350.pdf)]. We believe that such a highly technical analysis is beyond the scope of the current paper, and so is deferred to future work.
>
> **(Limitations And Societal Impact)[Suggestion 1.]** "The paper is overall easy to follow. In Corollary 1, I'll suggest the authors restate that the learning rate is set as the case (b) in Theorem 1. It confused me at first because it is not well-clarified."
>
> **[Response]** Sorry for the confusion caused. We will fix this problem in the camera-ready version.
>
> **(Limitations And Societal Impact)[Suggestion 2.]** "Furthermore, there's a series of works [2, 3] also providing finite-time analysis in the average reward setting using Actor-Critic with linear function approximation. However, all these works require the unique limit point of TD. Therefore, it would be interesting if the authors can comment about how the proposed method in this paper can be extended to these works to remove such uniqueness assumption. These comments will help readers understanding the contribution of this paper better."
>
> **[Response]** Indeed, [[Qiu et al., 2021](https://ieeexplore.ieee.org/stamp/stamp.jsp?arnumber=9435807&casa_token=1Zm72H39ztIAAAAA:yaq8teAJzlcrstprDKily4bSg8xzDpl_Gd16idCZXOx6SXXScLIakmw8G4aUk-Hvvx0EvSpk7g&tag=1); [Wu et al., 2020](https://arxiv.org/pdf/2005.01350.pdf)] assume the uniqueness of TD limit point, and such uniqueness does not hold in general, limiting the applicability of their results. As pointed out by the reviewer, an immediate future work is to overcome this limitation of these works. To remove the uniqueness assumption on the TD limit point in these works, one may consider the auxiliary algorithm proposed in part (1) of the proof of Theorem 1 (please see Appendix A.3 for details). The auxiliary algorithm overcomes the non-uniqueness by projecting each iterate $\theta_t$ onto the subspace $E$ (introduced in section 3.3, line 188). Thus, one obtains a unique limit point for the critic, that can be used in the actor. Note that this projection step can be easily implemented because it only needs the knowledge of the feature vectors, and does not need any knowledge of the underlying model. The auxiliary algorithm would also avoid the potential numerical instability of Algorithm 1. [[Qiu et al., 2021](https://ieeexplore.ieee.org/stamp/stamp.jsp?arnumber=9435807&casa_token=1Zm72H39ztIAAAAA:yaq8teAJzlcrstprDKily4bSg8xzDpl_Gd16idCZXOx6SXXScLIakmw8G4aUk-Hvvx0EvSpk7g&tag=1); [Wu et al., 2020](https://arxiv.org/pdf/2005.01350.pdf)] also introduce an artificial projection (onto a large enough ball) in the critic update to ensure that the iterates are bounded. Such a projection step is not practical in RL because one needs to know
> the problem parameters to pick the projection set so that the limit point lies in it. The auxiliary algorithm does not require this additional impractical projection step, because our analysis does not need the iterates to be bounded. Note that the projection in the auxiliary algorithm is completely different from the projection onto the ball in [[Qiu et al., 2021](https://ieeexplore.ieee.org/stamp/stamp.jsp?arnumber=9435807&casa_token=1Zm72H39ztIAAAAA:yaq8teAJzlcrstprDKily4bSg8xzDpl_Gd16idCZXOx6SXXScLIakmw8G4aUk-Hvvx0EvSpk7g&tag=1); [Wu et al., 2020](https://arxiv.org/pdf/2005.01350.pdf)].

---

> > ### Comment · Reviewer_ZKyN · 2021-08-24
> > **All concerns addressed**
> >
> > The reviewers have addressed all my concerns thus I would raise my score from 5 to 6

---

### Official Review · Reviewer_1neJ · 2021-07-16

**Rating:** 6
**Confidence:** 3

**Summary:**

This paper establishes the first finite-sample convergence results for average-reward TD(\lambda) policy evaluation with linear function approximation and average-reward tabular Q-learning in the synchronous setting.

**Limitations And Societal Impact:**

Yes

**Main Review:**

Strengths:
1. This work provides novel theoretical results for average-reward MDP with solid technical contributions;
2. The paper is well-written and the ideas are clearly explained.

I have several questions:

1. The current results are: policy evaluation with linear function approximation and tabular control. I wonder why the authors did not further extend to control with linear function approximation. Is there any technical challenge in achieving that?

2. The current convergence is to the best solution in the linear space i.e., \theta^*. I wonder whether the algorithm is robust to model misspecification, i.e., if the value function can not really be linearly represented, how the representation error (\|V-\Phi\theta^*\|) would affect the convergence.

3. In line 81, do you mean LSPE(\lambda) instead of LSTD(\lambda)?




**Time Spent Reviewing:**

10

---

> ### Author Response · Authors · 2021-08-10
> **Responses to Reviewer 1neJ**
>
> We thank the reviewer for the valuable feedback. Please find our responses below:
>
> **(Main Review)[Question 1.]**  "The current results are: policy evaluation with linear function approximation and tabular control. I wonder why the authors did not further extend to control with linear function approximation. Is there any technical challenge in achieving that?"
>
> **[Response]** We did not consider $Q$-learning with linear function approximation mainly because in general the algorithm is not stable and known to diverge due to the deadly triad [[Sutton \& Barto, 2018](https://scholar.google.com/citations?view_op=view_citation&hl=en&user=6m4wv6gAAAAJ&citation_for_view=6m4wv6gAAAAJ:IWHjjKOFINEC)] (i.e. off-policy learning, function approximation and bootstrapping). A simple example of divergence is Baird's counterexample [[Baird, 1995](https://citeseerx.ist.psu.edu/viewdoc/download?doi=10.1.1.50.7784&rep=rep1&type=pdf)]. There are works [[Melo et al., 2008](https://dl.acm.org/doi/pdf/10.1145/1390156.1390240?casa_token=0kY3bC_WpaYAAAAA:_nrRACGtUOdLShaFuA9DdUDUezVXD7YiYIMfCcsn0NdYrRzFl_MY8YsQ8ScGgvlWDEl2DpvMTFVCZQ); [Chen et al., 2019](https://arxiv.org/pdf/1905.11425.pdf)] that study convergence of $Q$-learning with linear function approximation in the discounted setting. To ensure the stability of the algorithm, they require a strong condition that involves the behavior policy, feature vectors, and the discount factor. The condition would often be violated when the discount factor is close to $1$ and the divergence of the algorithm is shown in the numerical experiments of [[Chen et al., 2019](https://arxiv.org/pdf/1905.11425.pdf)]. $Q$-learning under linear function approximation will need the development of a new algorithm that overcomes the deadly-triad. This is a major challenge, and is beyond the scope of the current paper.
>
> **(Main Review)[Question 2.]** "The current convergence is to the best solution in the linear space i.e., $\theta^*$. I wonder whether the algorithm is robust to model misspecification, i.e., if the value function can not really be linearly represented, how the representation error ($|V-\Phi\theta^*|$) would affect the convergence."
>
> **[Response]** We do not assume that the target differential value function lies in the linear function space (i.e. the column space of $\Phi$). Instead, we solve for a $\theta^*$  which is a solution of the projected Bellman equation (Eq. 3.4 in line 176). The convergence rate to one of the TD fixed point $\theta^*$ is *not* affected by the approximation error. However, model misspecification would cause large approximation error between the limit in the linear function space and the target differential value function of the form $v^{\mu} + ce$. As we are satisfied with an approximation of any differential value function, we define the approximation error, as the infimum of the weighted Euclidean distance from the set of differential value functions, i.e., $\inf_{c \in \mathbb{R}} \Vert \Phi \theta^* - (v^{\mu} + ce) \Vert_D$. The following approximation error bound holds [[Tsitsiklis \& Van Roy, 1999](https://web.stanford.edu/~bvr/pubs/average-td.pdf)]:
> $$\inf_{c \in \mathbb{R}} \Vert \Phi \theta^* - (v^{\mu} + ce) \Vert_D \leq \frac{1}{\sqrt{1-c_{\lambda}^2}} \inf_{\theta \in \mathbb{R}^d, c \in \mathbb{R}} \Vert \Phi \theta - (v^{\mu}+ce) \Vert_D,$$
> where the constant $c_{\lambda}$ is in $[0,1)$ for any $\lambda \in [0,1)$ and goes to $0$ as $\lambda \rightarrow 1$. Note that the term $\inf_{\theta \in \mathbb{R}^d, c \in \mathbb{R}} \Vert \Phi \theta - (v^{\mu}+ce) \Vert_D$ on the right-hand side is the minimal error possible given our approximation architecture, and becomes zero if there exist a vector $\theta$ and a scalar $c$ such that $\Phi \theta = v^{\mu} + ce$.
>
> **(Main Review)[Question 3.]** "In line 81, do you mean LSPE($\lambda$) instead of LSTD($\lambda$)?"
>
> **[Response]** Good catch on the typo. It should be LSPE($\lambda$) rather than LSTD($\lambda$). We will fix this in the camera-ready version.

---

### Official Review · Reviewer_bLoK · 2021-07-19

**Rating:** 6
**Confidence:** 3

**Summary:**

This paper focuses on the theoretical analysis (especially on the non-asymptotic sample efficiency) of two algorithms under the average-reward setting: (1) TD($\lambda$) with linear function approximation under Markovian observation noise (2) tabular Q-learning in the synchronous setting.

Previous works have obtained similar finite-sample guarantees under the discounted setting. However, this work focuses on the average-reward setting where the Bellman equation is known to have multiple fixed points. For the TD($\lambda$) analysis, the authors work in a subspace to get the unique solution of the projected Bellman equation. For the Q-Learning case, the Bellman operator is a contraction under the span seminorm.

**Limitations And Societal Impact:**

The authors addressed that one of the limitations of their work is that it is unclear whether Assumption 4 is necessary for establishing any finite time convergence bound.

**Main Review:**

The paper provides the first known finite-sample guarantees using both constant and diminishing step sizes of two widely applicable algorithms in RL.

There are, however, several points that can benefit the manuscript if tended to:

1)  It would be helpful to highlight the substantial differences in a nutshell from the present analysis and from [Zhang et al, 2021].

2) The paper proposed convergence guarantees for the TD learning algorithm and the Q-learning algorithm. However, currently, the sense of how tight the proposed convergence theorem and analysis are reflected in real/practical scenarios is missing.

The paper would greatly benefit from further theoretical/experimental discussions that investigate:

i)How does the change from discounted setting to the average-reward setting affect the terms in the main theorem (compared with [Zhang et al, 2021])?

ii) How does this algorithm converge in practice? In what zone does the theoretical analysis match the practical performance?

Overall, this paper provides insightful and novel theoretical results that broaden our analyzing tools on td learning and Q-learning under the stochastic approximation framework.

**Time Spent Reviewing:**

2

---

> ### Author Response · Authors · 2021-08-10
> **Responses to Reviewer bLoK**
>
> We thank the reviewer for the valuable feedback. Please find our responses below:
>
> **(Main Review)[Comment 1.]**  "It would be helpful to highlight the substantial differences in a nutshell from the present analysis and from [Zhang et al, 2021]."
>
> **[Response]**  [Zhang et al, 2021]  studies off-policy policy evaluation algorithm with linear function approximation, whereas we study on-policy algorithm. Off-policy TD with function approximation is known to diverge due to the deadly triad, and the focus of [Zhang et al, 2021] is to overcome that issue, which is very different from the focus of our paper. On-policy policy evaluation can be seen as a special case of off-policy policy evaluation. In the following, we take this viewpoint, and point out the differences with respect to  [Zhang et al, 2021].
>
> 1.  TD learning algorithm considered in our paper is a semi-gradient algorithm (incremental updates of TD are not stochastic gradient steps with respect to any fixed objective function) and is therefore difficult to show that it makes consistent, quantifiable, progress toward its asymptotic limit point. Our proof is based on the Lyapunov drift arguments and fast mixing of the underlying Markov chain. In contrast, Gradient TD (GTD) algorithm considered in [Zhang et al, 2021] is a stochastic gradient descent method for minimizing the mean squared projected Bellman error. In [Zhang et al, 2021], they formulate the original optimization as a primal-dual saddle point problem and leverage convergence analysis from that literature.
>
> 2. We do not require the uniqueness of the TD fixed point to guarantee the convergence of the iterates to one of the TD fixed points and establish the finite sample bound. On the contrary, Theorem 1 and Proposition 3 of [Zhang et al, 2021] require the Assumption 4.1 which guarantees the uniqueness of the TD fixed point. Theorem 2 and Proposition 4 of [Zhang et al, 2021] still need the uniqueness assumption for the algorithm to converge to the TD fixed point. Otherwise, the iterates of their algorithm may not converge to any TD fixed point due to their use of ridge regularization to ensure the stability of their algorithm.
>
> 3. We consider the non-i.i.d Markovian sample setting which is more practical than the i.i.d sample setting used in [Zhang et al, 2021].
>
> 4. To establish finite sample bounds of the proposed algorithms in Proposition 3 and Proposition 4, [Zhang et al, 2021] introduces an artificial projection (onto a large enough ball) to ensure that the iterates are bounded. Such a projection step is not practical in RL because one needs to know
> the problem parameters to pick the projection set so that the limit point lies in it. Our method does not require such a projection step.
>
> 5. We consider both constant and diminishing step sizes while [Zhang et al, 2021] only considers a constant step size in their finite sample analyses.
>
> **(Main Review)[Comment 2.]** "The paper proposed convergence guarantees for the TD learning algorithm and the Q-learning algorithm. However, currently, the sense of how tight the proposed convergence theorem and analysis are reflected in real/practical scenarios is missing."
>
> **[Response]** Our rates of convergence (i.e. $\tilde{\mathcal{O}}(\frac{1}{T})$ for TD and $\mathcal{O}(\frac{1}{T})$ for $Q$-learning) under the right choice of step-sizes match the state-of-the-art results in the discounted MDP setting. Moreover, mean-square error of $\mathcal{O}(\frac{1}{T})$ is the best one should expect for stochastic approximation type algorithms in general, because that is what one sees in the simplest setting of law of large numbers (and that is made formal by Cramer-Rao lower bound). To complement our theoretical results, we will include a numerical experiment section in the camera-ready version to demonstrate the tightness of our convergence bounds.
>
> **(Main Review)[Question i)]** "How does the change from discounted setting to the average-reward setting affect the terms in the main theorem (compared with [Zhang et al, 2021])?"
>
> **[Response]** From discounted setting to the average-reward setting, the rate of convergence with respect to time does not change. For example, the iterates of both discounted TD and average-reward TD converge with an $\tilde{\mathcal{O}}\left(\frac{1}{T}\right)$ rate using proper diminishing step sizes. However, the change of setting would affect other terms in front of $\tilde{\mathcal{O}}\left(\frac{1}{T}\right)$ in the convergence bounds. For example, when using decaying step sizes in the discounted TD, the term in front of $\tilde{\mathcal{O}}\left(\frac{1}{T}\right)$ is proportional to the effective horizon $1/(1-\tilde{\gamma})$, where $\tilde{\gamma}$ is the discount factor. The dependence on effective horizon plays a very important role in the discounted setting, because one can think of it as the time-horizon, over which the rewards are relevant (the rewards are too small beyond this horizon). In contrast, in the average-reward setting, there is no notion of the effective horizon, because we average the rewards over the entire infinite long horizon. We believe that a major advantage of our results is that the notion of the effective horizon vanishes.
>
> **(Main Review)[Question ii)]** "How does this algorithm converge in practice? In what zone does the theoretical analysis match the practical performance?"
>
> **[Response]** We will include a numerical experiment section in our camera-ready version to discuss the practical performance of our algorithms.

---

> > ### Comment · Reviewer_bLoK · 2021-08-27
> > **After Rebuttal Comment**
> >
> > Thanks for the detailed response! The authors have addressed all my concerns and I will keep my score.

---

### Decision · Program_Chairs · 2021-09-28

**Decision:**

Accept (Poster)

**Comment:**

This paper proves the sample complexity of $TD(\lambda)$ learning and Q-learning in the average reward setting. After the author response and reviewer discussion, the paper gathers enough support from the reviewers. Thus, I recommend acceptance.


**Consistency Experiment:**

NeurIPS has a long history of experimentation. In 2014, NeurIPS ran an experiment in which 10% of submissions were reviewed by two independent committees to quantify the randomness in the review process. This year, we repeated a variant of this experiment to see how the quality of the review process has changed over time.  This paper was part of the experiment and was therefore assigned to two committees (consisting of reviewers, an Area Chair, and a Senior Area Chair) that reached independent decisions.  If both committees made the same recommendation, this recommendation was followed. If a single committee recommended acceptance, the paper was accepted (with the exception of a few cases in which the other committee identified what we considered a fatal flaw, e.g., an error in a key result).

This copy’s committee reached the following decision: **Accept (Poster)**

The other committee assigned to the paper recommended **Reject**.  You can find the other set of reviews, along with any follow up discussion with the authors here:
https://openreview.net/forum?id=1Rxp-demAH0